# NeurIPS: Neuro-anatomical Inductive Priors for Sphere-based Brain Decoding

**Sijin Yu** [⋆ 1] **Zijiao Chen** [⋆ 2] **Zhenyu Yang** [1] **Zihao Tan** [1] **Jiakun Xu** [1] **Zhongliang Liu** [1] **Shengxian Chen** [1] **Wenxuan Wu** [3] **Xiangmin Xu** [4] **Xin Zhang** [† 1 5]

⋆equal contribution     †corresponding author

https://github.com/SCUT-Xinlab/NeurIPS

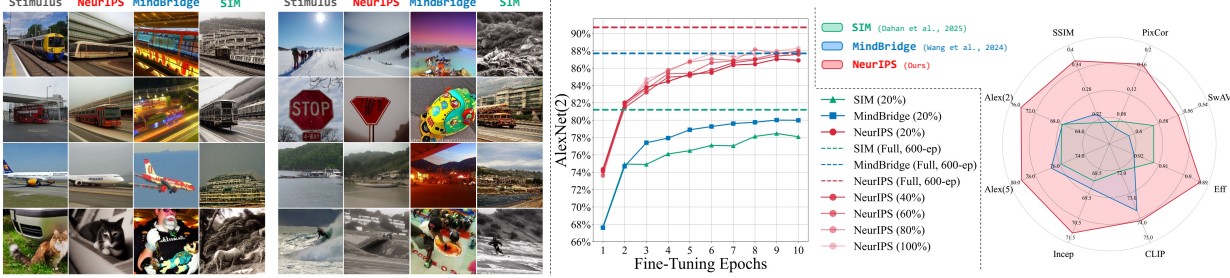

*Figure 1.* **NeurIPS achieves remarkable personalization efficiency.** NeurIPS adapts to a new user with minimal data and training. This figure shows the results for a held-out subject after personalizing a 3-subject, pre-trained model. The personalization is remarkably efficient, using only **20%** of the new subject's data for just **one epoch**. **(Left)** High-fidelity reconstructions after a single epoch demonstrate that the model quickly learns the new subject's neural patterns, already outperforming baselines. **(Middle & Right)** Performance metrics confirm the rapid adaptation. With only 20% of the data, the personalized model (solid lines) nearly matches the performance of a reference model trained on 100% of the data (dashed lines).

## Abstract

Current fMRI decoders face a performance-fidelity trade-off where efficient ID encoders outperform geometrically faithful surface-based models. We argue this is partly driven by inefficient surface tokenization and the failure to use anatomy as a predictive signal. We present **NeurIPS**, a framework that improves surface-based decoding by reframing anatomical variation from a nuisance to a powerful inductive prior. NeurIPS unites two innovations: a **Selective ROI Spherical Tokenizer (SRST)** for efficient geometric encoding, and a **Structure-Guided Mixture of Experts (SG-MoE)** that explicitly models individual anatomy using cortical features. On the Natural Scenes Dataset, NeurIPS establishes a new state-of-the-art for surface decoders and achieves performance comparable to strong 1D baselines. This is achieved with unprecedented efficiency, as the model converges dramatically faster (**10 vs. 600 epochs**). This efficiency enables rapid adaptation to new subjects using only **20%** of data and ensures robust scalability as the training cohort is expanded. Ablations provide causal evidence that these gains are driven by the model's use of cortical features, not by memorizing subject IDs. By leveraging anatomical priors, NeurIPS provides a principled and scalable path toward robust, generalizable brain decoding.

## 1. Introduction

> *"Variation is the hard reality, not a set of imperfect measures for a central tendency. Means and medians are the abstractions."*
>
> — Stephen Jay Gould, *The Median Isn't the Message* (1985)

What separates one mind from another? Is it noise to be averaged away, or is it the signal itself? We argue the latter: the precise brain geometry of our individual differences holds the key to mapping all minds onto a single, shared canvas. This question is central to brain decoding. Reconstructing images from fMRI has become a key proving ground for neural representation learning, with the goal of building systems that can generalize across people (Takagi & Nishimoto,

---

[1]South China University of Technology [2]Stanford University [3]King's College London [4]Foshan University [5]Pazhou Lab. Correspondence to: Xin Zhang <eexinzhang@scut.edu.cn>.

*Proceedings of the 43rd International Conference on Machine Learning*, Seoul, South Korea. PMLR 306, 2026. Copyright 2026 by the author(s).

2023; Chen et al., 2023; Scotti et al., 2023). Such generalization is not merely an academic benchmark; it is a prerequisite for a new generation of clinical and brain-computer interface applications that can be deployed robustly in the real world. However, the path to building such a universal decoder is obstructed by a significant challenge.

We believe the field's core obstacle should be viewed through the lens of *representation alignment*. The fundamental task is to map the functionally and anatomically unique cortical surface of each individual onto a common representational sphere, where the same stimulus evokes a consistent neural code. The current landscape reveals a deep divide. On one hand, computationally efficient 1D pipelines dominate benchmarks but achieve their speed by discarding the brain's native cortical geometry, thereby sacrificing the structural information crucial for principled alignment (Scotti et al., 2023; Huo et al., 2025). On the other hand, surface-based models, which preserve this geometry, have historically lagged in performance, creating what is often framed as an unavoidable performance-fidelity trade-off (Gu et al., 2023; Dahan et al., 2025; Yu et al., 2025). This trade-off, we argue, is not fundamental but a byproduct of two architectural mismatches that prevent effective alignment: (i) inefficient surface tokenization, and (ii) treating individual anatomical variation as noise.

These architectural mismatches manifest as specific limitations in prior work. Existing surface methods for fMRI (Gu et al., 2023; Yu et al., 2025) apply spherical convolutions across entire hemispheres. For static visual decoding tasks, this allocates significant computation to non-visual regions that may carry less stimulus-specific information, generating excessive tokens that destabilize model training. Similarly, prior cross-subject frameworks tackle anatomical variance inefficiently. They often condition their computations on subject IDs, which encourages the model to memorize individual patterns rather than learn generalizable rules about how structure shapes function. The root cause of these issues is a failure to leverage anatomical structure as a powerful inductive bias. A principled solution must therefore directly correct these flaws by adhering to two design principles: (1) efficient, geometry-aligned tokenization, and (2) anatomy-conditioned computation.

We implement these principles with two innovations, by introducing a hybrid architecture: a surface-based encoder that strictly operates on visual cortical ROIs to preserve geometry, followed by a compact latent transformer for efficient cross-subject modeling. The **Selective ROI Spherical Tokenizer** (SRST) directly addresses the first principle by confining spherical convolutions to visual ROIs, creating a stable, efficient token space that respects cortical geometry. The **Structure-Guided Mixture of Experts** (SG-MoE) implements the second principle by gating experts based on an individual's cortical thickness, curvature, and sulcal depth, rather than their ID, forcing specialization along meaningful structure-to-function axes. This anatomy-guided architecture provides the strong inductive bias that has been missing. Recent work on representation alignment (REPA) (Yu et al., 2024) has shown that such biases enable models to achieve strong performance in remarkably few epochs. This insight suggests that the slow convergence of existing methods is not fundamental but a consequence of poor inductive biases, a problem our architecture is designed to solve.

Additionally, rather than aiming for zero-shot generalization, we evaluate what matters for deployment: fast adaptation and scalability, demonstrating that anatomy-conditioned representations enable data-efficient transfer to new subjects. Our anatomy-guided architecture confirms that strong inductive biases unlock unprecedented learning speed. On the Natural Scenes Dataset (NSD) (Allen et al., 2022), our model adapts to a new subject by achieving 90% of its full performance with just 10 epochs of fine-tuning on only 20% of the subject's data. This represents a dramatic acceleration compared to conventional methods that require 200-600 epochs to converge (Wang et al., 2024). This rapid adaptation also translates to robust population-level scaling. As the training cohort grows, our model's performance remains stable while baselines that ignore anatomy degrade. This stability is a critical feature for building decoders that can be deployed reliably across diverse, real-world populations. Under matched compute, our model achieves comparable peak performance to competitive 1D pipelines while converging dramatically faster. A full suite of diagnostics and ablations confirms that these gains are driven by our two core principles: efficient, geometry-aligned tokenization and anatomy-conditioned computation.

**Contributions.** Our contributions are threefold.

**(A) Geometry-aware tokenization for efficient surface decoding.** We introduce SRST, an ROI-restricted spherical tokenizer that preserves cortical surface topology while reducing the practical token budget for visual decoding. This reduces memory and compute burden without claiming a change in the asymptotic attention complexity.

**(B) Anatomy-conditioned cross-subject modeling.** We introduce SG-MoE, which routes expert computation using cortical thickness, curvature, and sulcal depth instead of subject IDs. This provides a structured mechanism for modeling inter-subject variability, and our swap, random-anatomy, no-anatomy, and subject-ID ablations test whether the gains come from anatomy-conditioned routing rather than identity memorization.

**(C) Empirical validation at scale.** On NSD, NeurIPS achieves SOTA performance among surface-based decoders and comparable performance to strong 1D pipelines under

matched compute. It adapts efficiently to new subjects with limited data and remains stable when the training cohort is expanded. Ablations, routing analyses, and biological attribution maps further show that the model uses anatomical and visual-cortical structure in a systematic way.

## 2. Related Work

**Modern fMRI-to-Image Pipelines.** Reconstructing images from fMRI has rapidly advanced through two-stage pipelines: an fMRI encoder maps brain activity to a pre-trained latent space (e.g., CLIP, VAE), which then conditions a generative model, typically a diffusion model, for image synthesis (Takagi & Nishimoto, 2023; Chen et al., 2023; Shen et al., 2024). Recent work enhances reconstruction quality through multi-modal objectives (Scotti et al., 2023; 2024) or spatial controls (Huo et al., 2025). While these generative backends are powerful and computationally efficient, their fMRI encoders typically operate on flattened voxel vectors or 1D representations, meaning they do not explicitly model the underlying 2D cortical surface topology.

**Surface-based Decoders and the Efficiency Dilemma.** To preserve geometric fidelity, a growing family of decoders models brain activity directly on cortical manifolds. Spherical U-Net and its variants (Zhao et al., 2019; 2021) introduced spherical convolution and downsampling operations, showing how local neighborhoods can be preserved on spherical meshes. Gu et al. (2023) applied surface-based convolutional networks to natural-image decoding, demonstrating the value of cortical topology but using a relatively shallow surface-convolutional stack. Alternatively, SIM (Dahan et al., 2022; 2023; 2025) uses icosahedral surface patches and transformer-style modeling for multimodal decoding; while its patch grid preserves a surface coordinate system, the spatial selection is not task-adaptive to visual ROIs. Recently, Yu et al. (2025) incorporated surface-based fMRI and cortical structure, but their tokenizer is less tightly integrated with end-to-end cross-subject routing. In contrast, NeurIPS combines learnable visual-ROI spherical tokenization with anatomy-conditioned MoE routing in a single end-to-end cross-subject decoder.

**Inter-subject Variability and Registration.** Anatomical and functional variability across subjects is traditionally addressed using surface-based registration and multimodal alignment (Fischl, 2012; Robinson et al., 2014; 2018; Glasser et al., 2016b). These registration pipelines provide a common spherical coordinate system (e.g., *fsaverage*), but they do not eliminate residual subject-specific anatomical and functional variability. NeurIPS is built on this shared registration foundation. However, rather than treating the residual inter-subject variability merely as noise to be averaged away, our framework explicitly models it through anatomical routing.

**Subject Conditioning, MoE, and Memorization Risk.** To handle cross-subject decoding, several methods learn shared alignment spaces (Wang et al., 2024) or use subject embeddings, adapters, and ID-conditioned MoE routing (Quan et al., 2024). While these mechanisms are effective, they can scale poorly as cohorts grow and may encourage identity memorization rather than learning generalizable mapping rules. SG-MoE differs by using physical cortical features (thickness, curvature, sulcal depth, surface area) as routing signals. To test whether the performance gains stem from genuine anatomy-conditioned routing rather than implicit identity memorization, we explicitly evaluate subject-ID gating, swapped-anatomy, random-anatomy, and no-anatomy variants in our experiments.

## 3. Methodology

### 3.1. C-IB View of Cross-Subject Decoding

We frame cross-subject decoding via the Conditional Information Bottleneck (C-IB). The objective is to learn a representation $Z = T_\theta(X_s, A_s)$ that maximizes information about target $Y$ while suppressing subject-specific details, conditioned on anatomy $A_s$:

$$\max_\theta \ I(Z; Y \mid A_s) \ - \ \beta \, I(Z; \texttt{ID} \mid A_s). \qquad (1)$$

Our architecture approximates this objective implicitly. **SRST** maximizes the task-relevance term $I(Z; Y \mid A_s)$ by preserving topological geometry within visual ROIs. Simultaneously, **SG-MoE** minimizes the identity leakage term $I(Z; \texttt{ID} \mid A_s)$ by routing computation based on shared anatomical features rather than subject IDs, thereby learning generalizable mappings (see Appendix A.1).

### 3.2. Problem Definition & Alignment-Guided Principles

We study fMRI responses on the cortical surface within predefined visual ROIs (see Appendix Figure 9), registered to the standard FreeSurfer *fsaverage6* mesh (Fischl, 2012). Given NSD's static visual stimuli, we adopt a task-aligned strategy by restricting modeling to visual ROIs (consistent with Scotti et al. (2023); Wang et al. (2024); Yu et al. (2025)). This contrasts with full-cortex designs suited for multimodal paradigms (e.g., SIM (Dahan et al., 2025)); our analyses (Fig. 6D,E) confirm that signal contributions for this task concentrate in the visual hierarchy. For an image $y_{\texttt{img}}$, we utilize the GLM-estimated beta weights (which summarize the stimulus-locked BOLD response) as our input. The ROI-restricted beta maps on the left and right hemispheres are denoted as $x_{\texttt{L}} \in \mathbb{R}^{N_{\texttt{L}}}$ and $x_{\texttt{R}} \in \mathbb{R}^{N_{\texttt{R}}}$. We reconstruct $\hat{y}_{\texttt{img}}$ while conditioning on structural features $c_{s,\texttt{L}}$ and $c_{s,\texttt{R}}$ (thickness, area, sulcal depth, curvature), such that $\hat{y}_{\texttt{img}} = \mathcal{D}(x_{\texttt{L}}, x_{\texttt{R}} \mid c_{s,\texttt{L}}, c_{s,\texttt{R}})$. Our framework is guided by two principles designed to solve the alignment mismatches:

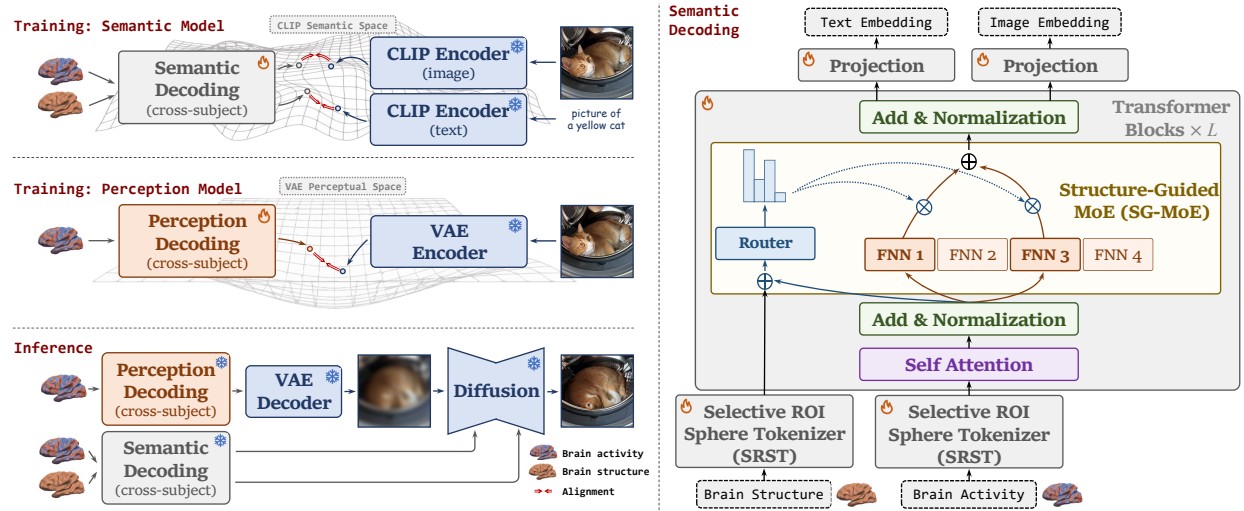

*Figure 2.* **The NeurIPS Framework. (Left)** Following established methods (Scotti et al., 2023), the overall pipeline is trained using a dual-decoder approach. The perception model maps fMRI to a VAE latent space for low-level detail, while the semantic model (aligns fMRI with a CLIP space for high-level content. During inference, both pathways jointly guide a frozen diffusion model to synthesize the final image. **(Right)** The core contributions of NeurIPS. A Selective ROI Spherical Tokenizer (SRST) first efficiently extracts features from both brain activity and anatomical structure. These are then processed by a transformer backbone where the standard feed-forward network is replaced by our Structure-Guided Mixture of Experts (SG-MoE), which uses anatomical information to route tokens to specialized experts for improved cross-subject generalization.

**Principle 1: Efficient, Topology-Preserving Tokenization for Geometric Alignment.** Surface/spherical operators preserve cortical neighborhoods and provide the correct inductive bias for alignment (Bronstein et al., 2017; Cohen et al., 2018). Importantly, SRST does not change the $\mathcal{O}(T^2)$ asymptotic complexity of transformer attention. Instead, it reduces the token count $T$ by restricting surface modeling to visual ROIs. For *fsaverage6*, full-cortex processing involves 40,962 vertices per hemisphere (81,924 total), whereas the NSD visual ROI contains 9,488 vertices (4,613 left / 4,875 right), corresponding to an 88.4% reduction. This substantially reduces the practical memory and compute burden, yielding a compact, topology-preserving space suitable for joint training. The visual-ROI mask is a strong task-specific prior tailored to static visual decoding.

**Principle 2: Anatomy as a Conditional Prior for Functional Alignment.** Neuroscientific evidence shows anatomy predicts function, from V1 retinotopy to ventral stream organization (Engel et al., 1997; Dumoulin & Wandell, 2008; Weiner et al., 2014; Natu et al., 2021). Instead of using subject IDs (Quan et al., 2024), we condition expert computation on anatomical features $c_s$. This approximates the conditional distribution $p(Y \mid X_s, A_s)$ and reduces inter-subject heterogeneity, enabling generalizable functional alignment.

### 3.3. The NeurIPS Framework Overview

To instantiate these principles, our **NeurIPS** framework (Figure 2) uses a dual-decoder architecture to align fMRI signals with target representations. The pipeline contains two decoding branches: a semantic decoder $\mathcal{D}_\text{S}$ that maps fMRI to a CLIP space (§3.4), and a perceptual decoder $\mathcal{D}_\text{P}$ that maps fMRI to a VAE space (§3.5). We integrate our two innovations within the critical semantic path: the Selective ROI Spherical Tokenizer (SRST) to solve the topology mismatch (§3.4), and the Structure-Guided Mixture of Experts (SG-MoE) to solve the identity mismatch (§3.4). Finally, the outputs from both decoders are combined to steer a versatile diffusion model for high-quality image reconstruction (§3.6). To further validate the generalizability of our learned representations, we also present results on secondary tasks such as brain captioning in Appendix Table 6 and Appendix Figure 10.

**Information Flow.** The overall pipeline proceeds as follows (Fig. 2): First, **SRST** extracts geometry-aware tokens from the visual cortex (Fig. 3). These tokens are processed by the transformer backbone, where **SG-MoE** dynamically routes information based on anatomical structure. The resulting representations simultaneously drive the Semantic Decoder (aligning with CLIP) and the Perception Decoder (aligning with VAE). Finally, both outputs guide a frozen diffusion model to generate the reconstruction.

### 3.4. Semantic Path: SRST + SG-MoE

The semantic model $\mathcal{D}_\text{S}$ maps fMRI signals to the semantic space of CLIP (Radford et al., 2021). For a given fMRI pair $(x_\text{L}, x_\text{R})$ and corresponding subject-specific structure features $(c_{s,\text{L}}, c_{s,\text{R}})$, $\mathcal{D}_\text{S}$ outputs the predicted CLIP image and text embeddings by $\hat{e}_\text{image}, \hat{e}_\text{text} = \mathcal{D}(x_\text{L}, x_\text{R} | c_{s,\text{L}}, c_{s,\text{R}})$.

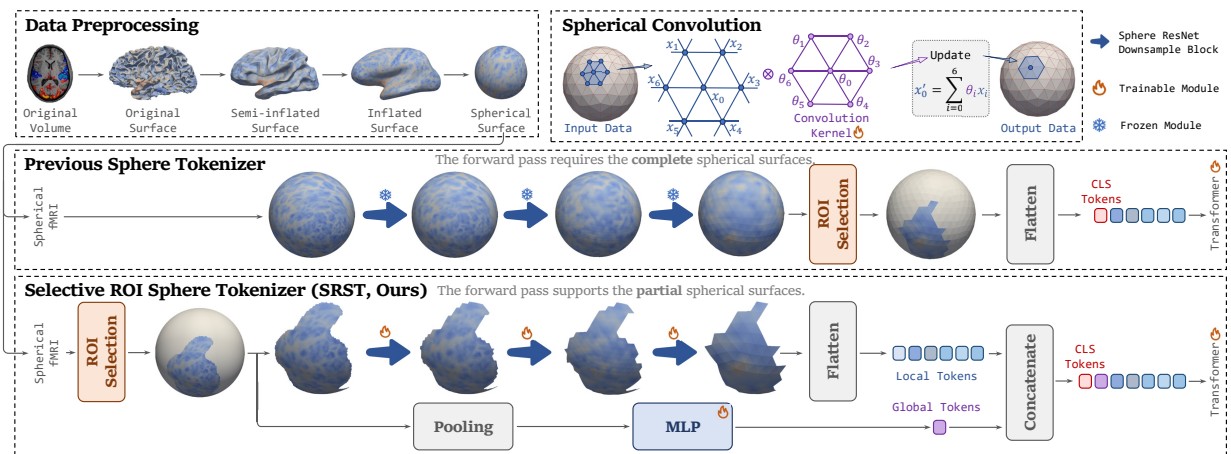

*Figure 3.* **Selective ROI Spherical Tokenizer (SRST) design and efficiency gains.** The standard pipeline maps volumetric fMRI signals to a spherical surface where hexagonal convolution kernels can preserve local topology. However, prior methods inefficiently processed all 40,962×2 hemisphere vertices (Yu et al., 2025). In contrast, our SRST restricts computation to only the 9,488 vertices within visual ROIs, yielding an 88.4% reduction in processed surface points, while preserving local topology. From these selected vertices, SRST generates both **spatially-detailed local tokens** and a **semantically-rich global token**. This efficiency is critical, making end-to-end training of a deep transformer on surface-based fMRI data both practical and stable.

The ground truth $e_{\texttt{image}}$ and $e_{\texttt{text}}$ are derived by passing the image and its caption into the CLIP encoders $\mathcal{E}_{\texttt{image}}$ and $\mathcal{E}_{\texttt{text}}$, respectively. The model $\mathcal{D}_{\texttt{S}}$ is then optimized using an MSE loss: $\mathcal{L} = ||e_{\texttt{image}} - \hat{e}_{\texttt{image}}||_2^2 + ||e_{\texttt{text}} - \hat{e}_{\texttt{text}}||_2^2$.

**Selective ROI Spherical Tokenizer (SRST).** Our SRST implements this geometric efficiency principle through selective computation. As illustrated in Figure 3, while prior methods apply spherical convolutions uniformly across the hemisphere, SRST adopts a task-aligned approach, performing computation on vertices within visual ROIs. For the NSD benchmark, this selective approach reduces the token budget by 88.7%, creating a compact token space suitable for stable joint-optimization with a transformer backbone. To capture information at multiple scales, SRST generates two complementary representations: **local tokens** from the localized convolutions to preserve fine-grained geometric patterns, and **global tokens** via flattening and pooling to provide overall scene context. This dual-token design provides the transformer with both high-fidelity spatial details and high-level semantic information without resorting to destructive flattening. The global token is obtained after geometry-aware spherical processing and pooling; it is therefore not equivalent to flattening raw cortical activity before surface modeling.

**Structure-Guided Mixture of Experts (SG-MoE).** While prior frameworks often use subject embeddings or adapters, these scale linearly with cohort size and risk overfitting. Our SG-MoE replaces identity-based routing with anatomy-conditioned expert selection. By gating experts based on local cortical features rather than subject IDs, we force the model to learn generalizable structure-to-function rules

shared across individuals, rather than memorizing subject-specific patterns. A standard Mixture of Experts (MoE) replaces a transformer's feed-forward network (FFN) with $N$ parallel expert FFNs, and a router network $\mathcal{R}$ selects a sparse subset of these experts for each token. Instead of conditioning this routing on a learned subject ID embedding as in prior work (Quan et al., 2024), we implement a purely structure-guided router. The gating network receives only local cortical features (thickness, curvature, sulcal depth, and location descriptors) as input; it does not receive any subject-ID tokens or embeddings. This ensures that the MoE experts specialize based solely on local cortical geometry rather than memorizing subject identity. The four cortical features ($c_s$) are first mapped into a structural embedding $e_{s,\texttt{stru}}$ via a lightweight SRST tokenizer. The routing decision for an input token $e$ is then modified to be $w = \mathcal{R}(e|e_{s,\texttt{stru}})$. We implement the SG-MoE based on DeepSeek-V3 (Liu et al., 2024; Guo et al., 2025) MoE framework, with $N = 16$ experts per layer and a top-$k$ routing strategy ($k = 6$). This design forces experts to specialize based on structural properties, allowing NeurIPS to learn a generalizable structure-to-function mapping. To test whether these structural inputs act as meaningful anatomical priors rather than implicit subject identifiers, we evaluate subject-ID gating, swapped anatomy, random anatomy, and no-anatomy variants in §4.4.

### 3.5. Perception Path

The perceptual decoding model, $\mathcal{D}_{\texttt{P}}$, aims to project fMRI responses into the latent perceptual space of a Variational Autoencoder (VAE) (Kingma et al., 2013). Unlike the semantic path, this auxiliary branch prioritizes low-level

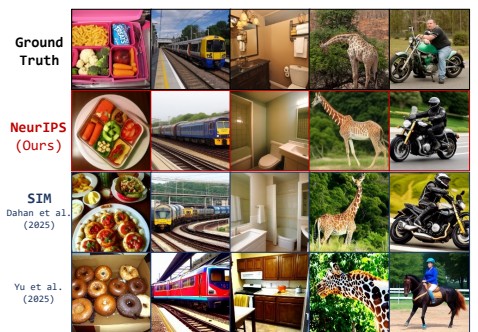
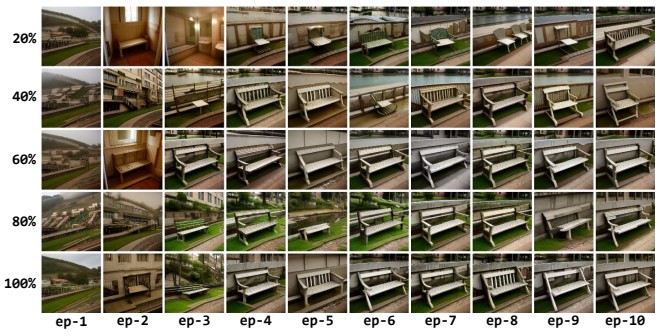
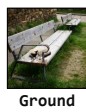

*Figure 4.* NeurIPS demonstrates superior reconstruction quality and rapid new-subject adaptation on the NSD `test` set. **(Left)** Qualitative comparison on the standard **within-subject benchmark**, where each model is trained on a single subject's full dataset. NeurIPS reconstructions show higher fidelity to object identity, layout, and fine details compared to prior surface-based models. **(Right)** Demonstration of **fast new-subject adaptation**. The model is pretrained on a cohort of subjects and then fine-tuned on a held-out subject using a limited data budget. Rows correspond to the percentage of fine-tuning data (20-100%), and columns represent the number of training epochs (1-10). Even with minimal exposure (**one epoch on 20% of data**), the model generates a semantically coherent reconstruction, with quality progressively improving as the data and training budget increase.

visual fidelity over geometric interpretability. Therefore, we follow Scotti et al. (2023) and flatten the ROI-masked fMRI signals into a 1D vector input, treating this stream as a standard MLP-based mapping. Given an input fMRI pair $(x_\text{L}, x_\text{R})$, the model predicts the corresponding VAE latent code $\hat{z} = \mathcal{D}_\text{P}(x_\text{L}, x_\text{R})$. We use the same pre-trained VAE as Stable Diffusion (Rombach et al., 2022). The model $\mathcal{D}_\text{P}$ is then optimized using a standard MSE loss: $\mathcal{L}(z, \hat{z}) = ||z - \hat{z}||_2^2$. We do not claim the perceptual path as a geometry-aware contribution. Our main architectural innovations reside in the semantic path, where cortical topology and cross-subject anatomical variation are directly modeled. Extending the perceptual branch to spherical-domain processing is an important direction for future work.

### 3.6. Image Reconstruction

The final image reconstruction is performed by a pre-trained, versatile diffusion model (Xu et al., 2023). The model is guided by the outputs of both the perceptual and semantic decoders. The predicted VAE latent $\hat{z}$ from the perceptual decoder $\mathcal{D}_\text{P}$ provides low-level structural information. The predicted CLIP text embeddings $\hat{e}_\text{text}$ and image embeddings $\hat{e}_\text{image}$ from the semantic decoder $\mathcal{D}_\text{S}$ provides high-level semantic guidance. Both latents are injected into the diffusion model's U-Net architecture via separate cross-attention layers, allowing the model to synthesize an image $\hat{y}_\text{img}$ that is faithful to both the perceptual details and the semantic content of the original stimulus.

## 4. Experiments and Results

**Dataset and Preprocessing.** We utilize the Natural Scenes Dataset (NSD) (Allen et al., 2022), a large-scale fMRI–image paired benchmark. Four subjects (subj01, 02, 05, 07) completed the full protocol, viewing 10,000 images

from COCO (Lin et al., 2014) with three repetitions each. Among these, 1,000 images were shared by all subjects and are designated as the common `test` set. The remaining are partitioned into 8,500 for `train` and 500 for `val`. For scalability experiments, `train` data from four additional subjects (subj03, 04, 06, 08) are included in the training set.

**Evaluation Metrics.** Following standard evaluation protocols, we assess reconstruction quality using 8 metrics. For low-level visual fidelity, we use pixel-wise correlation (`PixCor`) and structural similarity (`SSIM`). For feature-level similarity, we use 2-way comparisons on activations from AlexNet's 2nd and 5th layer (`Alex(2)`, `Alex(5)`) (Krizhevsky et al., 2012), InceptionV3 (`Incep`) (Szegedy et al., 2016), and CLIP ViT-L/14 (`CLIP`) (Radford et al., 2021). Last, we measure average correlation distance using EfficientNet-B1 (`Eff`) (Tan & Le, 2019) and SwAV-ResNet50 (`SwAV`) (Caron et al., 2020).

**Baseline Configuration and Parity.** To ensure a fair comparison, we re-implemented the surface-based baselines (SIM and Yu et al. (2025)) using their official code or reported settings, aligning loss functions, data splits, and optimization schedules with our method. For SIM (Dahan et al., 2025), we specifically scaled its transformer to match our model's width and depth, reporting this stronger variant in Table 2. Crucially, **all reconstruction results** in this paper, including those for MindBridge, SIM, and Yu et al., were generated using the exact same Versatile Diffusion backend and identical inference hyperparameters (50 steps, guidance scale 7.5, text-image mixup 0.5). FreeSurfer-based surface reconstruction and spherical projection require approximately 42 hours as a one-time preprocessing step per subject. After preprocessing, model inference takes approximately 3.4 seconds per volume on a single NVIDIA A800 GPU. During training, NeurIPS requires ∼61.5GB VRAM

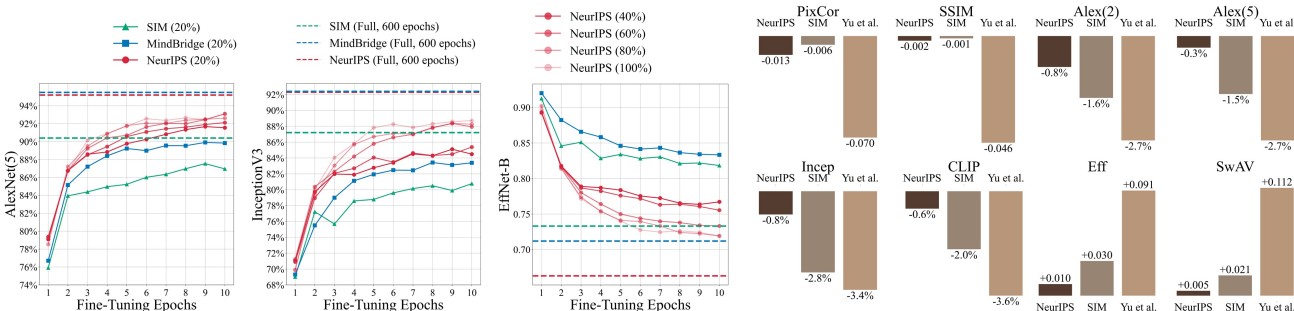

*Figure 5.* **NeurIPS achieves rapid new-subject adaptation and robust scalability.** In all panels, models were pretrained on 3 subjects and then fine-tuned on a held-out subject. We compared our results with prior models (SIM (Dahan et al., 2025), MindBridge (Wang et al., 2024), and Yu et al. (2025)). **(Left)** The adaptation curves plot performance over 10 fine-tuning epochs. With just 20% of the new subject's data (solid lines), NeurIPS (red) consistently outperforms baselines and rapidly approaches its asymptotic performance (dashed line). **(Right)** The bar plot illustrates scalability by showing the performance degradation when the training cohort is expanded from 4 to 8 subjects. Taller bars indicate a larger drop (note: Eff/SwAV signs are inverted for consistency). NeurIPS shows the smallest degradation across 7 out of 8 metrics, confirming its superior robustness.

*Table 1.* Quantitative comparison of cross-subject performance on the NSD (Allen et al., 2022) test set. All methods were trained as a single model on four subjects 01, 02, 05, 07, and the results shown are averaged across the shared test set. Performance is evaluated across eight standard metrics, with the best result in each category (1D-vector and Sphere-based) marked in bold. The Input size column clarifies the input dimensionality, distinguishing between ROI voxels for 1D methods and surface tokens for sphere-based methods (where "×2" denotes both hemispheres). The results show that our model, NeurIPS, establishes a new state-of-the-art for surface-based decoders and achieves performance parity with top-tier 1D pipelines on high-level semantic metrics. See related analysis in §4.1.

| Method | Trained Voxels | Low-Level | | | | High-Level | | | |
|---|---|---|---|---|---|---|---|---|---|
| | | PixCor↑ | SSIM↑ | Alex(2)↑ | Alex(5)↑ | Incep↑ | CLIP↑ | Eff↓ | SwAV↓ |
| ***1D-vector-based fMRI Methods*** | | | | | | | | | |
| Mind-Vis (Chen et al., 2023) | 13811 | 0.067 | 0.196 | 67.7% | 74.2% | 67.9% | 69.3% | 0.898 | 0.513 |
| Takagi & Nishimoto (2023) | 13811 | 0.246 | **0.410** | 78.9% | 85.6% | 83.8% | 82.1% | 0.811 | 0.504 |
| MindEye (Scotti et al., 2023) | 13811 | 0.129 | 0.255 | 84.2% | 89.2% | 84.1% | 85.0% | 0.812 | 0.487 |
| MindBridge (Wang et al., 2024) | 13811 | 0.151 | 0.263 | 87.7% | 95.5% | 92.4% | **94.7%** | 0.712 | 0.418 |
| UMBRAE (Xia et al., 2025) | 13811 | **0.283** | 0.328 | **93.9%** | **96.7%** | **93.4%** | 94.1% | **0.700** | **0.393** |
| NeuroPictor (Huo et al., 2025) | 13811 | 0.141 | 0.349 | 91.4% | 95.7% | 88.3% | 88.9% | 0.722 | 0.417 |
| ***Sphere-based fMRI Methods*** | | | | | | | | | |
| Gu et al. (2023) | 32492×2 | 0.103 | 0.264 | - | - | - | - | 0.892 | 0.508 |
| Yu et al. (2025) | 9548 | 0.165 | 0.305 | 78.2% | 89.0% | 85.1% | 88.3% | 0.733 | 0.398 |
| SIM (Dahan et al., 2025) | 40962×2 | 0.119 | 0.260 | 81.2% | 90.4% | 87.2% | 89.4% | 0.733 | 0.448 |
| **NeurIPS (w/o preception decoding)** | 9488 | 0.148 | 0.283 | 86.7% | 94.5% | 92.2% | 92.9% | **0.662** | **0.396** |
| **NeurIPS (Ours)** | 9488 | **0.248** | **0.370** | **90.7%** | **95.2%** | **92.3%** | **93.2%** | 0.663 | 0.404 |

and ∼138 seconds per epoch. Although NeurIPS has more total parameters due to the MoE experts, only a sparse subset is activated per token. More importantly, its new-subject adaptation converges within 10 fine-tuning epochs, substantially reducing the total wall-clock training budget compared to baselines.

### 4.1. SOTA Geometry- and Anatomy-Aware Decoders

To answer this, we present a comprehensive quantitative comparison in Table 1. The results show that NeurIPS attains a new state-of-the-art among surface-based decoders across all 8 metrics. For instance, on the high-level CLIP metric, NeurIPS achieves 93.2%, significantly outperforming the prior surface model, SIM (89.4%). Critically, NeurIPS substantially narrows the gap between surface-

based and strong 1D pipelines, especially on high-level semantic metrics. While computationally intensive 1D models like MindBridge reach a CLIP score of 94.7%, our model's 93.2% achieves comparable performance under matched compute, demonstrating that incorporating explicit cortical geometry reduces the apparent performance-fidelity trade-off. This quantitative strength is supported by qualitative results in Figure 4 (Left), where NeurIPS reconstructions better preserve object identity, layout, and fine-grained texture compared to prior surface models. We provide further qualitative results in Appendix Figure 13.

### 4.2. NeurIPS Efficiently Adapts to New Subjects

NeurIPS achieves substantially faster new-subject adaptation. To test this, after multi-subject pretraining, we fine-

*Table 2.* Ablation studies on the NSD (Allen et al., 2022) `test` set providing causal evidence for our architectural design. Each row modifies a single component relative to the full model. Detailed analysis is provided in Section 4.4.

| # | Setting | Low-Level | | | | High-Level | | | |
|---|---------|-----------|---|---|---|------------|---|---|---|
| | | PixCor↑ | SSIM↑ | Alex(2)↑ | Alex(5)↑ | Incep↑ | CLIP↑ | Eff↓ | SwAV↓ |
| 1 | w/o global token | 0.236 | 0.370 | 87.7% | 92.7% | 87.7% | 89.4% | 0.723 | 0.441 |
| 2 | subject ID gating | 0.247 | 0.370 | 90.2% | 94.9% | 92.0% | 92.7% | 0.668 | 0.407 |
| 3 | w/o perception decoding | 0.148 | 0.283 | 86.7% | 94.5% | 92.2% | 92.9% | **0.662** | **0.396** |
| 4 | w/o semantic decoding | **0.369** | **0.512** | 69.5% | 65.4% | 53.3% | 55.0% | 1.004 | 0.651 |
| 5 | Yu-style structure fusion | 0.239 | 0.359 | 89.7% | 94.2% | 90.9% | 91.7% | 0.669 | 0.412 |
| 6 | full brain | 0.193 | 0.316 | 85.4% | 91.7% | 89.4% | 91.0% | 0.723 | 0.443 |
| 7 | functional features gating | 0.241 | 0.371 | 90.1% | 94.9% | 91.7% | 92.7% | 0.661 | 0.397 |
| 8 | w/o left brain tokens | 0.151 | 0.288 | 85.2% | 93.3% | 90.1% | 91.5% | 0.691 | 0.414 |
| 9 | w/o right brain tokens | 0.153 | 0.289 | 85.2% | 93.5% | 90.2% | 91.6% | 0.688 | 0.414 |
| 10 | shuffle spherical position | 0.159 | 0.292 | 86.5% | 93.3% | 90.6% | 91.5% | 0.686 | 0.419 |
| 11 | convolution receptive field = 1 | 0.160 | 0.292 | 87.1% | 93.6% | 90.4% | 91.7% | 0.682 | 0.416 |
| 12 | anatomical swap | 0.242 | 0.366 | 89.4% | 94.4% | 90.9% | 91.9% | 0.666 | 0.411 |
| 13 | no anatomy | 0.240 | 0.361 | 88.1% | 93.4% | 90.8% | 90.2% | 0.676 | 0.417 |
| 14 | random anatomy | 0.244 | 0.369 | 89.9% | 94.4% | 91.2% | 92.0% | 0.666 | 0.409 |
| 15 | **Full Model (Correct Anatomy)** | 0.248 | 0.370 | **90.7%** | **95.2%** | **92.3%** | **93.2%** | 0.663 | 0.404 |

tune the model on a held-out subject (subj01) using only a fraction of their available data. The adaptation curves in Figure 5 show that with just 20% of the new subject's data, NeurIPS (red curve) rapidly approaches its asymptotic performance within 10 fine-tuning epochs. After just one epoch on this limited dataset, NeurIPS already produces a semantically coherent reconstruction (Figure 4, Right). We additionally evaluate a more constrained 5% setting. With only 5% of the held-out subject's data and 10 fine-tuning epochs, NeurIPS still reaches 86.0% `Alex(5)` and 80.0% `CLIP`, indicating that the anatomical prior remains useful under severe data limitations (see Appendix Table 8). Appendix Table 8 provides a comprehensive set of qualitative examples illustrating this progressive improvement.

### 4.3. Performance Remains Stable as the Cohort Scales

Our model's anatomy-conditioned routing provides superior scalability. To assess this, we compare performance when training on four subjects versus all eight, evaluating on the same four-subject test set. The results (Figure 5, Right) reveals superior robustness: when the training cohort expands from 4 to 8 subjects, SIM's CLIP score drops by 2.0 points, whereas NeurIPS drops by only 0.6 points. This suggests that anatomy-conditioned routing effectively treats increased subject variability as signal rather than noise. Our model's stability confirms that it effectively leverages this variability, enabling it to generalize robustly to larger, more realistic population sizes. For a complete, per-subject breakdown, see Appendix Tables 4 and 5.

### 4.4. Ablation Studies

To isolate component contributions, we conducted the ablation studies in Table 2. **Geometric Validity and Efficiency**. Disrupting topology via spherical shuffling (#10) or restricting receptive fields to 1 (#11) impairs performance, confirming SRST leverages cortical neighborhoods rather than simple feature pooling. A full-cortex tokenizer (#6) increases memory cost while lowering CLIP scores, validating our task-aligned ROI efficiency. **Dual Decoders**. Removing the semantic (#4) or perception (#3) decoders leads to collapse in their respective metrics, proving both are indispensable. To explicitly test whether SG-MoE learns meaningful structure-to-function mappings rather than implicitly memorizing subject identities, we conducted targeted anatomy routing ablations (Table 2). Correct anatomical routing achieves the best performance. Replacing anatomy with subject IDs (#2), swapped anatomy from other subjects (#12), random anatomical features (#14), or no anatomical input (#13) consistently decreases overall performance, especially on high-level semantic metrics. Because these variants keep the MoE capacity fixed and only modify the routing signal, these systematic (though modest) differences suggest that the gains are driven by valid anatomical structure rather than solely parameter count or identity memorization.

### 4.5. Neuroscientific Plausibility

Our analyses reveal that NeurIPS's cross-subject alignment stems from its ability to operate on a geometrically appropriate coordinate system and to condition on the true source of inter-subject variability. An analysis of the SG-MoE router (Figure 6A) shows that expert selection exhibits high dependence on a token's cortical origin but minimal dependence

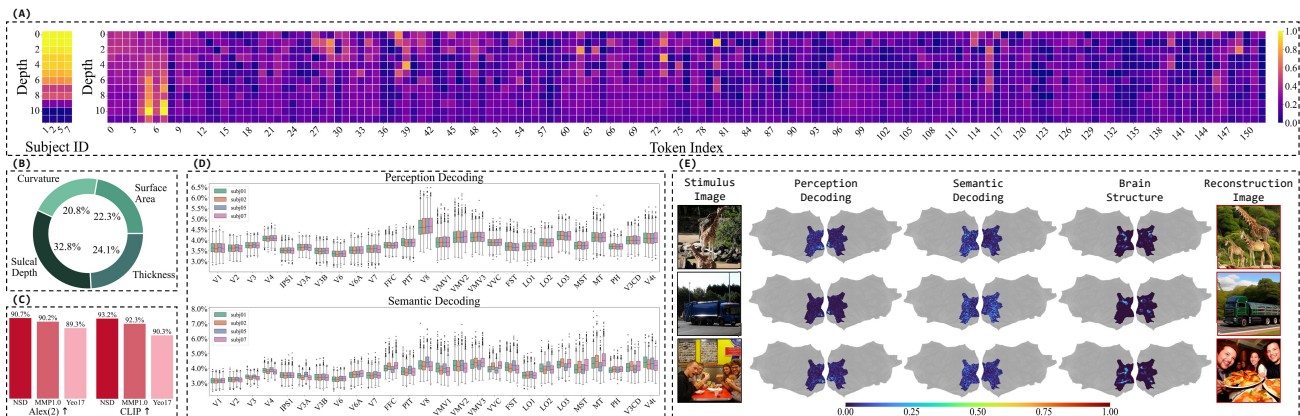

*Figure 6.* **NeurIPS learns anatomy-aware representations on a geometry-preserving basis. (A)** Expert routing dependence maps show that computation is driven by a token's cortical location (high *region* dependence, right) rather than the subject's identity (low *subject* dependence, left). Special tokens are indicated: `[CLS]`=0–3, `[global]`=4–7. **(B)** Structural feature attributions confirm a complementary use of all four anatomical features, ruling out single-feature shortcuts. **(C)** The geometry-aligned tokenization of SRST is validated, as the NSD visual-ROI scheme outperforms generic atlases on key high-level metrics. **(D)** An ROI-wise analysis reproduces the brain's visual hierarchy across all subjects, with performance improving from early (V1-V3) to higher-order ventral areas. **(E)** Surface contribution maps confirm that decoding activity is concentrated in the visual cortex, with semantic decoding extending further into the ventral stream than perception decoding.

on the subject's identity. This demonstrates a progressive disentanglement of information: subject-specific signals are suppressed from task-general tokens (`[CLS]`) with network depth, while being intentionally carried by the dedicated `[global]` tokens (See Appendix §C for detailed discussion). Furthermore, feature attributions (Figure 6B) confirm that the router systematically relies on a complementary set of anatomical cues, a finding that is consistent across all subjects (see Appendix Figure 11 for individual results). This rules out identity memorization or single-feature shortcuts. Furthermore, we find that subject pairs with more similar anatomical features tend to exhibit more similar routing patterns (see Appendix Table 9). This further supports the interpretation that the router responds to physical anatomical similarity rather than merely memorizing subject IDs.

The representations learned via this anatomy-aware mechanism are neuroscientifically plausible. An ROI-wise performance analysis (Figure 6D) reproduces the known visual hierarchy across all subjects, with decoding accuracy improving from early visual areas (V1-V3) toward the ventral stream. Furthermore, contribution maps (Figure 6E) confirm that the model's decoding activity is correctly concentrated in the visual cortex. Crucially, we also validate our foundational design choice. The superiority of our task-focused, topology-preserving tokenizer over generic atlases (Figure 6C) provides direct evidence that an appropriate geometric basis is critical for effective alignment (see Appendix Figure 12 for full results across all metrics). Together, these findings provide a mechanistic explanation for our model's empirical success: a stable geometric basis from SRST combined with anatomy-aware computation from SG-MoE directly enables the rapid new-subject adaptation and

robust scalability demonstrated in our results.

## 5. Discussion and Conclusion

We show that the performance-fidelity trade-off is an artifact of inefficient tokenization and ignored anatomical variance. SRST enables efficient surface training, while SG-MoE ensures cross-subject generalization by conditioning on anatomy. Together, they match strong 1D baselines with superior scalability, evidenced by robust cohort expansion and rapid adaptation. We propose this geometry-anatomy synergy as a robust recipe for future neuro-AI models.

Regarding scope, while visual ROIs maximize efficiency for NSD, full-cortex modeling remains preferable for multimodal paradigms engaging distributed networks. We also acknowledge dependencies on registration quality. Ethically, we rely exclusively on consented data and advocate for strong safeguards against non-consensual inference as this technology matures.

## Impact Statement

This work involves human participants and may therefore raise potential ethical considerations related to data collection, privacy protection, and responsible use. All data employed in this study are derived from the NSD dataset (Allen et al., 2022). The authors of the NSD dataset have provided detailed ethical statements, indicating that the data collection procedure was approved under appropriate ethical oversight and that all participants gave informed consent for the use of their physiological and neuroimaging data in scientific research. Since our study only uses the publicly released data and does not involve additional data collection or direct interaction with participants, the potential risks to participants are further minimized.

In addition, we open-source NeurIPS to support transparency, reproducibility, and future research in the community. To encourage responsible use, we explicitly require that all users refrain from employing the released resources for unethical, harmful, or illegal purposes, including but not limited to privacy infringement, unauthorized identification of individuals, or any application that may negatively affect human participants.

## Acknowledgement

This work is funded by General Program of the National Natural Science Foundation of China (NSFC-62471185), Key-Area Research and Development Program of Guangdong Province (2023B0303040001), Guangdong Basic and Applied Basic Research Foundation (2024A1515010180, 2025A1515012836), and Guangdong Provincial Key Laboratory of Human Digital Twin (2022B1212010004).

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

# A. An Information-Theoretic and Anatomical View of NeurIPS

The relationship between artificial intelligence (AI) and neuroscience has become a highly productive area of research. The "black-box" nature of both deep neural networks and the human brain has motivated using AI frameworks to model and interpret neurological processes. Such efforts include capturing spatial encoding in the hippocampus (Whittington et al., 2021; Kim et al., 2023; Ellwood, 2024), replicating semantic representations (Huth et al., 2016; Millet et al., 2022; Caucheteux et al., 2023; Antonello & Huth, 2024), and reproducing visual representations in the cortex (Wen et al., 2018; Ozcelik & VanRullen, 2023; Benchetrit et al., 2023; Tang et al., 2023). Here, we provide a theoretical framework to explain why our proposed architecture, NeurIPS, is inherently better suited for this task. We argue that the benefits of our two main contributions: the Selective ROI Spherical Tokenizer (SRST) and the Structure-Guided Mixture of Experts (SG-MoE), can be rigorously understood through the lens of information theory and neuroanatomy.

## A.1. A Conditional Information Bottleneck Framework for Decoding

Let $X_s$ denote the ROI-restricted fMRI signals on the cortical surface of subject $s$ (registered to a common spherical chart), $A_s$ be the corresponding anatomical fields (e.g., cortical thickness, area, sulcal depth, and curvature), and $Y$ be the target representation we aim to align with (e.g., SD-VAE latents or CLIP embeddings (Kingma et al., 2013; Radford et al., 2021)). A decoder is a transformation $T$ that maps the brain data to a compressed representation $Z = T(X_s, A_s)$.

The goal of cross-subject decoding can be formalized as a **Conditional Information Bottleneck (C-IB)** (Tishby et al., 2000; Zhuang et al., 2025) problem. We seek a representation $Z$ that is maximally informative about the target $Y$ while being minimally informative about the subject's identity (ID), all conditioned on the known anatomical features $A_s$. This can be expressed as:

$$\max_T \; I(Z; Y \mid A_s) \; - \; \beta \, I(Z; \texttt{ID} \mid A_s)$$

Here, $I(Z; Y \mid A_s)$ is the task-relevant information we want to preserve, while $I(Z; \texttt{ID} \mid A_s)$ is the subject-specific "nuisance" information we want to suppress. The hyperparameter $\beta$ controls this trade-off.

## A.2. Why Surface Representations Are Informationally Superior to 1D Flattening

The superiority of surface-based methods stems from how they handle the inherent geometry of the cortex.

**The Information Cost of Flattening.** According to the Data-Processing Inequality (DPI), any preprocessing step $X_s \to \tilde{X}_s$ cannot increase information, i.e., $I(\tilde{X}_s; Y) \leq I(X_s; Y)$. Flattening fMRI data into a 1D vector is a destructive form of preprocessing that discards explicit geodesic neighborhoods and sulcal topology. This overlooks the meaningful spatial auto-correlation of signals across the cortex (Margulies et al., 2016; Bijsterbosch et al., 2018; Kong et al., 2019; Shinn et al., 2023; Leech et al., 2023) and forces the network to implicitly relearn these fundamental spatial relationships from the data.

**SRST as an Efficient Information-Preserving Transformation.** In contrast, our SRST is designed to be a more efficient information processor. This is motivated by evidence that the spatial connectivity features of the cortex are key to its function (Smith et al., 2013; Glasser et al., 2016a; Vidaurre et al., 2017; Pervaiz et al., 2022), making surface modeling a more accurate representation that has long been employed in cortical analysis (Glasser et al., 2016b; Margulies et al., 2016; Gordon et al., 2017a;b). By restricting computation to task-relevant visual ROIs, SRST removes a significant source of noise (non-visual brain signals), thereby improving information *efficiency*. Crucially, by using spherical convolutions, it explicitly preserves the local surface neighborhoods, retaining the geometric information that 1D flattening discards.

## A.3. Why Anatomy-Conditioning (SG-MoE) Is Superior to ID-Conditioning

The core of cross-subject generalization lies in how a model handles inter-subject variability.

**ID-Conditioning as Memorization.** Prior MoE models that condition on a subject's ID (Quan et al., 2024) essentially learn a mixture of subject-specific experts. This encourages the model to *memorize* "who" a subject is, which directly increases the nuisance information term $I(Z; \texttt{ID})$.

**Anatomy-Conditioning as Principled Generalization.** Our SG-MoE instead conditions the expert routing on anatomical features $A_s$. This leverages the well-established empirical link between cortical morphology and function (e.g., retinotopic organization follows sulcal patterns (Engel et al., 1997; Natu et al., 2021)). By doing so, SG-MoE approximates the conditional distribution $p(Y \mid X_s, A_s)$ rather than just $p(Y \mid X_s)$. This allows the model to "explain away" the portion of

variance in $X_s$ that is attributable to anatomy $A_s$. This directly reduces the remaining subject-specific information, lowering $I(Z; \text{ID} \mid A_s)$ and leading to a more generalizable representation. The expert selection for a token $X_s^{(i)}$ with corresponding anatomy $A_s^{(i)}$ can be modeled as:

$$Z^{(i)} = \sum_{k=1}^{K} \pi_k(X_s^{(i)}, A_s^{(i)}) \cdot \text{Expert}_k(X_s^{(i)})$$

where the gating weights $\pi_k$ depend **exclusively** on the anatomical structure $A_s^{(i)}$, ensuring that routing decisions generalize across subjects with similar local geometry.

### A.4. Empirical Confirmation of Theoretical Predictions

Our theoretical framework leads to several testable predictions, all of which are confirmed by our experiments in the main paper.

1. **Prediction (Efficiency):** SRST's ROI restriction should reduce computational load without sacrificing performance. **Confirmation:** This is validated in §4.4. Our ablation (Table 2, #6) shows that a full-cortex tokenizer consumes significantly more memory (74GB vs 61GB) while achieving lower accuracy than our visual-ROI SRST.

2. **Prediction (Fast Personalization):** A model that learns generalizable structure-function rules should adapt to a new subject's anatomy much faster. **Confirmation:** This is shown in §4.3, where NeurIPS achieves high-fidelity results with just 20% of a new subject's data in a few epochs.

3. **Prediction (Scaling Robustness):** A model that explains away anatomical variance should be more stable when the diversity of the training cohort increases. **Confirmation:** This is demonstrated in §4.3, where NeurIPS exhibits the smallest performance degradation when scaling from 4 to 8 training subjects.

4. **Prediction (Anatomy Usage):** The SG-MoE router should genuinely use anatomical information, and removing this information should harm performance. **Confirmation:** This is proven by multiple ablations (Table 2): replacing anatomy with **subject ID** (#2) or **functional statistics** (#7) both degrade performance. Furthermore, feature attribution analysis (Fig. 6) confirms systematic reliance on sulcal depth and curvature.

## B. How NeurIPS Differs from Prior Surface Models

NeurIPS integrates several key architectural innovations that distinguish it from prior surface-based decoders, moving from static, full-hemisphere processing to a learnable, anatomy-aware, and ROI-restricted framework. We summarize the differences in spherical operations in Table 3.

*Table 3.* Comparison of spherical operations across surface-based fMRI decoders.

| Method | Spherical Operation | Kernel/Projection | Spatial Selection | Training Status |
|---|---|---|---|---|
| Yu et al. (2025) | SphericalUNet-style (Zhao et al., 2019) | Learned | Full Hemisphere | Pretrained & Frozen |
| SIM (Dahan et al., 2025) | Icosahedral Patches | Learned Linear Proj. | Fixed Grid (Unlearnable) | Learned End-to-End |
| **NeurIPS (Ours)** | SphericalUNet-style (Zhao et al., 2019) | Learned | Visual ROI (Task-Aligned) | **Learned End-to-End** |

### B.1. Tokenization: From Brute-Force to Selective and Learnable

A major bottleneck in prior work has been inefficient tokenization. While prior surface decoders utilize learned operations, they are often constrained by fixed grids or frozen weights.

**Learnable vs. Fixed Spherical Operations.** It is important to clarify the distinction between learned kernels and fixed heuristics in prior work:

- **Yu et al. (Yu et al., 2025)** employ a ResNet-style spherical downsampling architecture with SphericalUNet-style (Zhao et al., 2019) convolutions. While their convolutional kernels are indeed result of optimization (learned) rather than hand-crafted filters, these layers are typically pretrained and subsequently *frozen* during the fMRI-to-image training phase.

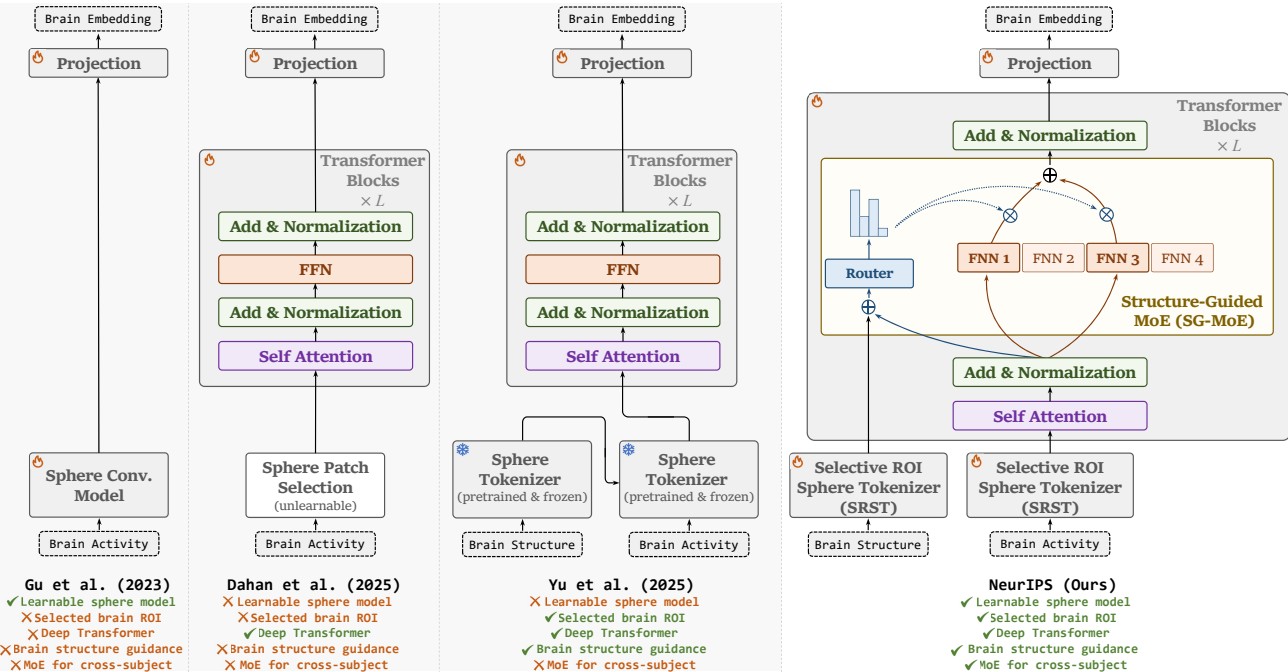

*Figure 7.* **Comparison with other surface-based fMRI decoders.** This figure contrasts prior methods with our proposed **NeurIPS** framework. Early approaches like Gu et al. (2023) used a shallow spherical convolutional stack on the full hemisphere. Subsequent work incorporated deep transformers but with key limitations: SIM (Dahan et al., 2025) selected surface patches with a fixed, non-learnable heuristic, while Yu et al. (2025) used frozen, pretrained tokenizers. These models either ignored anatomical structure or used it only as a static input feature. **NeurIPS** introduces two key advances: (1) its Selective ROI Sphere Tokenizers (SRST) are fully learnable and restrict computation to relevant visual areas, and (2) its Structure-Guided MoE (SG-MoE) uses cortical anatomy to dynamically route information. This end-to-end, anatomy-aware design makes surface-based decoding computationally efficient and robust to cross-subject differences.

- **SIM (Dahan et al., 2025)** uses a regular icosahedral tessellation to define a grid of surface patches. While it applies a learned linear projection within each patch, the patch selection itself is based on a fixed grid. We describe this as "unlearnable" because the grid locations are predefined and not adapted to the task; the patch encoder is trainable, but the spatial selection is not.

**The NeurIPS Approach.** In contrast, NeurIPS solves the issues of frozen representations and excessive token counts. SRST uses SphericalUNet-style (Zhao et al., 2019) convolutions following Yu et al. (2025) but learns all kernels *end-to-end* from scratch on the NSD dataset. Furthermore, rather than processing the entire hemisphere (which incurs an $\mathcal{O}(T^2)$ complexity burden) (Yu et al., 2025) or using a fixed grid (Dahan et al., 2025), SRST restricts computation to functionally defined visual ROIs. This makes the tokenization process both fully learnable and computationally efficient.

### B.2. Use of Anatomy: From Static Feature to Dynamic Routing

Previous models have either ignored cortical anatomy or included it as just another static feature concatenated with brain activity. This fails to capture how anatomical differences shape functional responses. Our **Structure-Guided MoE (SG-MoE)** represents a paradigm shift, using an individual's anatomical features (e.g., sulcal depth, cortical thickness) to dynamically route information through different expert pathways. This allows the model to learn a generalizable structure-to-function mapping, rather than simply memorizing subject-specific patterns.

### B.3. Synthesis: An End-to-End, Anatomy-Aware System

In summary, NeurIPS is the first surface-based decoder to combine end-to-end learnable, ROI-restricted tokenization with dynamic, anatomy-conditioned routing in a deep transformer. This principled design directly addresses the core challenges of computational efficiency and cross-subject heterogeneity, making surface-based decoding practically scalable and competitive with top-tier 1D pipelines.

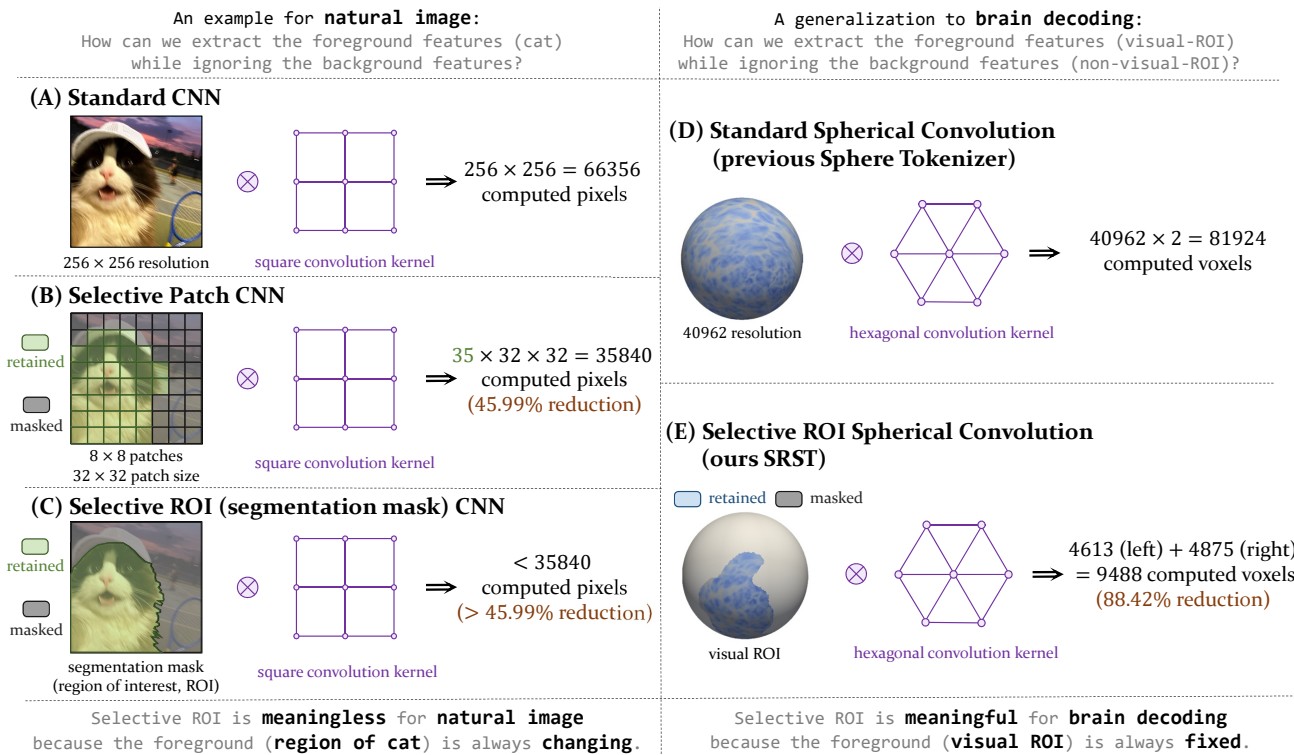

*Figure 8.* Efficiency of Selective ROI Spherical Convolution. We contrast standard CNNs (A) and Spherical Convolutions (D) with our proposed selective approach. While selective masking is difficult in natural images due to shifting foregrounds (C), it is highly effective in brain decoding where the visual cortex is anatomically consistent. By applying spherical convolutions exclusively to the visual ROI (E), our SRST achieves an **88.42% reduction** in computational cost compared to processing the full cortical mesh, focusing the model capacity solely on task-relevant features.

## C. An Analysis of MoE: What Determines Experts Routing?

Our analysis of the model reveals that two main factors influence experts routing: the ***brain region*** (i.e., the model selects experts based on which brain region the current token belongs to) and the ***subject*** (i.e., the model selects experts based on which subject the current token corresponds to). In order to examine the dependence of experts routing on various factors, we compute the activated experts count of the $i$-th token at the $d$-th layer of the Transformer for every subject and every NSD test sample ($\forall i, \forall d$). The results are represented as a vector of length $N$, with each element indicating the activation count for the corresponding expert. Here, $N$ denotes the total number of experts (as defined in §3.4). We ultimately obtain an array count $\in \mathbb{N}^{4 \times \texttt{depth} \times T \times N}$ to record the expert activation for each token in the Transformer for the 4 subjects.

**Experts Routing Dependence on Brain Region.** To study the degree of experts selection dependence on brain regions, we compute the variance of the statistical results at the token level (ignoring [CLS] and [global] tokens). Positions with higher variance indicate greater differences in experts activation across tokens, suggesting a higher dependency of experts selection on the brain region at those positions. As shown on the Figure 6A, the dependence on brain regions is higher in the shallower layers (closer to the input). As the model progressively extracts and integrates embeddings, the reliance on brain regions decreases in the deeper layers (closer to the output).

**Experts Routing Dependence on Subject.** Similarly, we compute the variance of the statistical results across the 4 subjects to measure the extent to which expert routing is influenced by different subjects, as shown on the Figure 6A. **(1)** For the [CLS] tokens, we find that experts selection exhibits slight dependence on the subject in the shallower layers, but almost no subject dependence in the deeper layers. This is because the [CLS] tokens in the final layer directly aligns with the CLIP embeddings, which is a subject-independent target. **(2)** For the [global] tokens, we observe a significant subject dependence in its experts routing overall, as the [global] tokens aggregates the complete brain information from fMRI. Moreover, the subject dependence of the [global] tokens increases layer by layer, whereas the subject dependence of regular tokens decreases across layers. This indicates that the [global] tokens serves to capture subject-specific

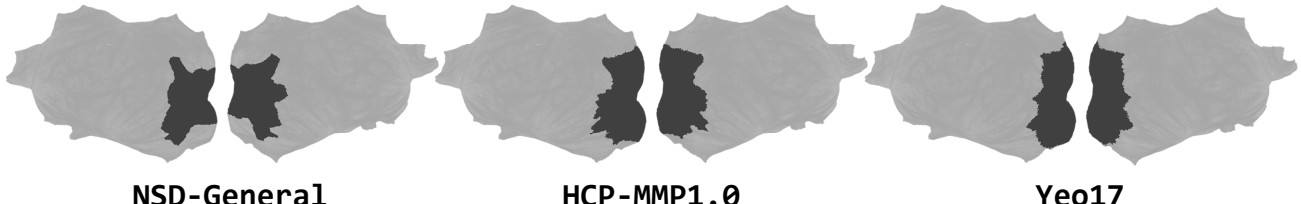

*Figure 9.* Schematic diagrams of visual ROIs for three different parcellations.

information. In the cross-subject brain decoding task, as the model deepens, the subject specificity of the [CLS] tokens and regular tokens is gradually eliminated to better generalize across subjects, while subject-specific information is transferred to the [global] tokens via the attention mechanism. This observation supports the rationale of our model design, demonstrating its effectiveness.

## D. Implementation Details

**Model Frameworks.** For perception decoding (§3.5), we fully adopt the MindEye (Scotti et al., 2023) approach, using our ROI-masked fMRI (i.e., voxels within the visual brain region) as a 1D vector input to the network. For semantic decoding (§3.4), we employ the frozen pretrained CLIP model, specifically OpenAI's ViT-L/14. In the SRST (§3.4), the functional module gradually downsamples the fMRI from the *fsaverage6* space to the *fsaverage3* space with the resolution at each layer being $[64, 128, 256, 512]$, while the structural module downsamples the spherical brain structure from the *fsaverage6* space to the *fsaverage1* space with the resolution at each layer being $[16, 32, 64, 128, 256, 512]$. In the SG-MoE (§3.4), the implementation of the MoE module is based on DeepSeek-V3 (Liu et al., 2024; Guo et al., 2025). Our model consists of $N = 16$ routed experts and 2 shared experts, with the number of activated experts set to 6 and the intermediate dimension set to 512. The Transformer has an embedding dimension of 768, a depth of 12 layers, and 12 attention heads. Dropout is applied in the sphere tokenizer with a probability of 0.3 and in the attention mechanism with a probability of 0.5.

**Training Strategy.** For perception decoding (§3.5), we train with a total batch size of 64 (in the 4-subject training case, 16 samples per subject per iteration). The model is trained for a total of 100 epochs. For semantic decoding (§3.4), the total batch size is 96 (with a batch size of 24 per subject in the 4-subject training case). The model is trained for a total of 600 epochs. For both perception and semantic decoding, the learning rate is set to $10^{-4}$, weight decay is 0.01, and the maximum gradient norm is 0.1. The same settings are used for fine-tuning. All experiments are conducted on an 80GB Nvidia A800 GPU.

**Inference Strategy.** We employ Versatile Diffusion (VD) (Xu et al., 2023) for image reconstruction. Let the prediction of perception decoding (§3.6) be $z_{\text{fMRI}}$. We mix it with Gaussian noise $\varepsilon \sim \mathcal{N}(\mathbf{0}, \mathbf{1})$ at intensity $t$ to obtain the initial noise $z$ for diffusion: $z = t \cdot z_{\text{fMRI}} + (1 - t) \cdot \varepsilon$. We set $t = 0.1$. The number of inference steps is set to 50, the text-image mixup ratio in VD is 0.5, and the classifier-free guidance scale is set to 7.5.

**ROI Parcellation.** We use a total of 3 ROI parcellations: NSD-General (Allen et al., 2022), HCP-MMP1.0 (Glasser et al., 2016a), and Yeo17 (Yeo et al., 2015). An analysis of the influence of different parcellations on brain decoding performance is provided in §4.5 and Figure 6. NSD-General is the visual brain region ROI provided by the official NSD dataset (Allen et al., 2022), consisting of 9,488 visual voxels (4,613 for left, 4,875 for right). NSD-General is used by default for all other experiments. For HCP-MMP1.0 (Glasser et al., 2016a), we select the following brain regions as visual ROIs, resulting in a final ROI containing 10,192 voxels (5,147 for left, 5,045 for right).



V1,V2,V3,V3A,V3B,V3CD,V4,V4t,V6,V6A,V7,V8,
IPS1,FFC,PIT,VMV1,VMV2,VMV3,VVC,FST,LO1,LO2,LO3,MST,MT,PH



For Yeo17 (Yeo et al., 2015), we chose Visual A and Visual B as visual ROIs, yielding a final ROI consisting of 9,128 voxels (4,549 for left, 4,579 for right). In Figure 9, we present the visual brain regions corresponding to the three different parcellations.

**Brain Heatmaps and Importance Analysis.** Figure 6 presents an analysis of the contribution and importance of the input data, conducted using Grad-CAM (Selvaraju et al., 2017).

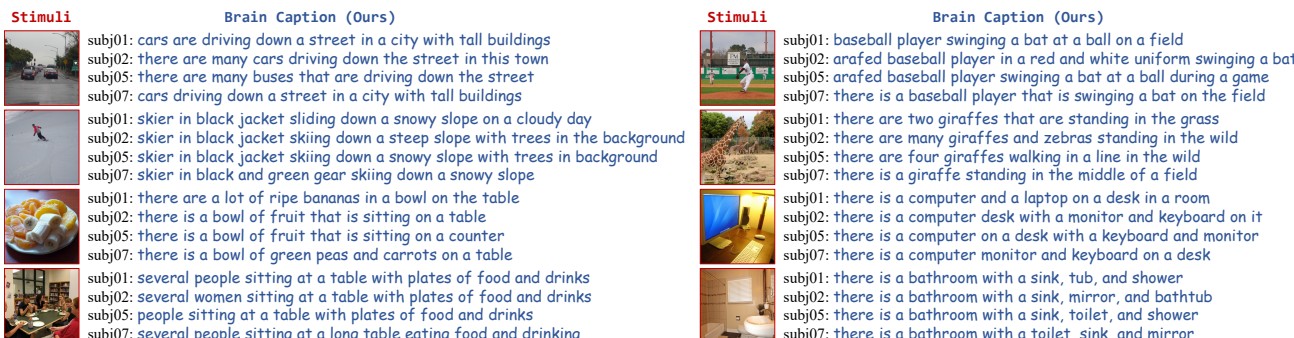

*Figure 10.* Qualitative examples of brain captioning. Our model accurately describes visual stimuli for different subjects, validating its generalization ability in the brain captioning task.

## E. More Results

**Brain Captioning.** Although not our primary objective, we follow UMBRAE (Xia et al., 2025) and conducted brain captioning experiments. Quantitative comparisons with sphere-based baselines are reported in Table 6, while qualitative examples of brain captioning are shown in Figure 10, demonstrating both the effectiveness of our model and its adaptability to different tasks.

**Brain Retrieval.** We follow MindEye's (Scotti et al., 2023) pipeline for the brain retrieval task, with quantitative results reported in Table 7. The results of the retrieval task corroborate and support our claims alongside the outcomes of the image reconstruction task.

**Comparison to Previous Work.** We provide a detailed quantitative comparison against state-of-the-art baselines in Table 1. Since the original SIM paper (Dahan et al., 2025) did not conduct training on the NSD dataset (Allen et al., 2022), we rigorously reproduced their method using the official open-source code. We trained it on NSD from scratch and report the corresponding metrics. To ensure a strictly fair comparison, we aligned all other components—including loss functions, training strategies, and the inference pipeline—to be identical to our NeurIPS framework. Furthermore, noting that the original SIM employed a relatively small Transformer backbone, we scaled up its hyperparameters (width and depth) to match our model capacity. We observed that this scaled version performed better than the original configuration, so we report this stronger "Improved SIM" variant in our tables to provide a competitive baseline. Similarly, since the code for Yu et al. (2025) is not open-source, we meticulously reproduced their work based on the architectural details and guidelines provided in their paper. The results in Table 1 are reported as averages across subjects in the multi-subject setting. For a more granular view, detailed metrics for individual subjects are provided in Table 4 for the 4-subject training case and in Table 5 for the 8-subject scalability experiment. Crucially, to eliminate any confounding factors from the generative model, all reconstructions presented in this paper—including those for MindBridge (Wang et al., 2024), SIM (Dahan et al., 2025), Yu et al. (2025), and NeurIPS (Figure 1, Figure 4)—were generated using the exact same Versatile Diffusion backend and identical inference hyperparameters (see Appendix D). Therefore, the performance gaps observed in Table 1 and the visual quality differences in our figures can be attributed solely to the quality of the fMRI encoder representations.

**More Reconstruction Results.** We provide additional qualitative results to demonstrate the robustness of our model across diverse semantic categories. In Figure 4 of the main text, we highlighted the superior reconstruction quality of NeurIPS compared to sphere-based baselines. Here, Figure 13 expands on this by showcasing randomly selected reconstructions for multiple subjects. NeurIPS exhibits strong cross-subject consistency, faithfully reconstructing complex scenes involving animals (e.g., zebras, bears), human activities (e.g., skiing, baseball), and indoor settings (e.g., kitchens, bathrooms). The model captures not only the high-level semantics but also fine-grained details such as object orientation, texture, and background elements, further validating the efficacy of our geometry-aware tokenization.

**More New Subject Adaption Fine-tuning Results.** A key advantage of our anatomy-guided architecture is its ability to rapidly adapt to new individuals with minimal data. To rigorously assess this, we conducted extensive fine-tuning experiments simulating a "new user" scenario with limited data and compute budgets. Figure 1 in the main text displays the impressive reconstruction results after fine-tuning for just one epoch on 20% of the data. Figure 5 quantifies this rapid adaptation, showing steep learning curves that quickly approach asymptotic performance. For a more comprehensive visual analysis, Figure 14 presents a grid of reconstructions under varying constraints (y-axis: 20%-100% data; x-axis: 1-10

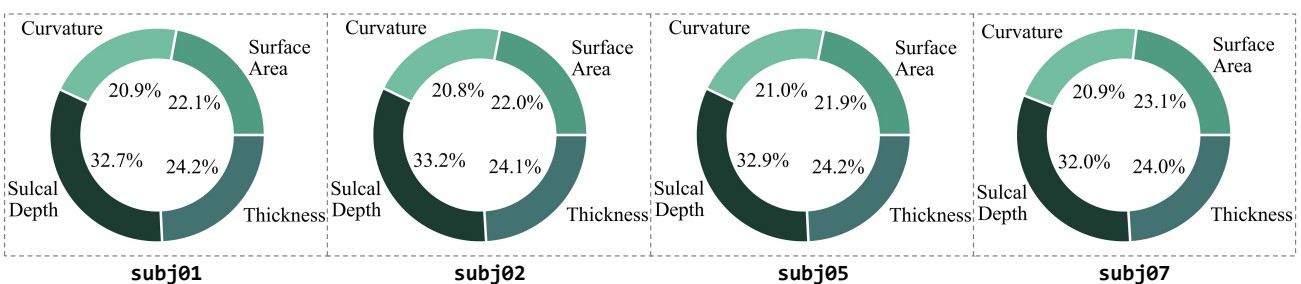

*Figure 11.* More analysis of cortical structural feature importance for different subjects.

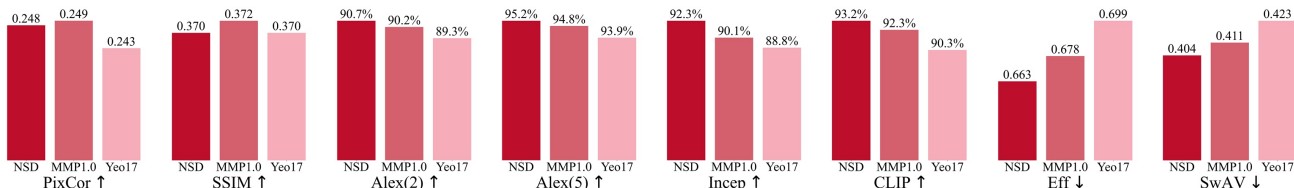

*Figure 12.* **Validation of the visual ROI parcellation scheme.** This figure compares the performance of our model when using three different visual Regions of Interest (ROIs) as the basis for the tokenizer. We evaluate our primary scheme, the functionally-defined NSD-General mask, against two standard anatomical atlases: HCP-MMP1.0 and Yeo17. The results show that the NSD-General parcellation consistently yields superior performance across all eight evaluation metrics. This validates our design choice and indicates that a task-aligned, functionally-defined ROI basis is more effective for image reconstruction than generic anatomical atlases.

epochs). We observe that even under the most stringent constraint (20% data, 1 epoch), NeurIPS produces semantically coherent images that capture the gist of the stimulus. As data and training time increase, the reconstructions progressively refine, recovering sharper details and more accurate textures. This confirms that our model effectively leverages the pre-learned structure-function mapping to accelerate personalization.

In addition to the qualitative results, we provide a quantitative breakdown of the extreme low-data regime in Table 8. Even with only 5% of the target subject's data, the model achieves strong semantic alignment (80.0% CLIP), indicating that the anatomical prior is highly effective when data is severely restricted.

**More Brain Structure Importance Results.** To verify that our SG-MoE router genuinely utilizes anatomical information, we analyzed the feature attribution scores for the gating mechanism. In Figure 6B of the main text, we presented the importance scores for Subject 1. Figure 11 extends this analysis to all four subjects in the training cohort. Consistently across all subjects, we observe that the router relies on a balanced combination of all four anatomical features—sulcal depth, curvature, cortical thickness, and surface area—rather than overfitting to a single metric. This consistency across individuals strongly supports our claim that the model has learned a generalizable, anatomy-based rule for routing information, rather than memorizing subject-specific identities.

**Routing Similarity vs. Anatomical Similarity.** To further validate that the SG-MoE router relies on physical anatomical structure rather than merely acting as an implicit identity memorizer, we computed the pairwise anatomical similarity and routing decision similarity between held-out subjects. As shown in Table 9, subject pairs with higher anatomical similarity (e.g., Subj01 and Subj05) tend to exhibit more similar expert routing patterns. This correlation provides supplementary evidence that the router's behavior is genuinely governed by cortical geometry.

*Table 4.* Quantitative results for each subject on NSD `test` (4 subjects training).

| Subject | Method | Low-Level | | | | High-Level | | | |
|---------|--------|-----------|---|---|---|------------|---|---|---|
| | | PixCor↑ | SSIM↑ | Alex(2)↑ | Alex(5)↑ | Incep↑ | CLIP↑ | Eff↓ | SwAV↓ |
| subj01 | Yu et al. (2025) | 0.172 | 0.314 | 78.6% | 88.7% | 84.8% | 88.9% | 0.736 | 0.396 |
| subj01 | SIM (Dahan et al., 2025) | 0.125 | 0.262 | 82.0% | 91.0% | 88.2% | 89.7% | 0.728 | 0.447 |
| subj01 | **NeurIPS (Ours)** | 0.271 | 0.375 | 91.8% | 95.8% | 93.0% | 93.8% | 0.656 | 0.398 |
| subj02 | Yu et al. (2025) | 0.167 | 0.302 | 77.7% | 89.0% | 85.9% | 88.2% | 0.733 | 0.394 |
| subj02 | SIM (Dahan et al., 2025) | 0.121 | 0.262 | 81.7% | 90.9% | 86.9% | 88.7% | 0.735 | 0.449 |
| subj02 | **NeurIPS (Ours)** | 0.256 | 0.373 | 91.0% | 94.9% | 91.7% | 92.2% | 0.672 | 0.407 |
| subj05 | Yu et al. (2025) | 0.163 | 0.305 | 78.6% | 90.1% | 86.4% | 89.6% | 0.717 | 0.393 |
| subj05 | SIM (Dahan et al., 2025) | 0.119 | 0.262 | 81.9% | 91.0% | 88.6% | 90.8% | 0.720 | 0.437 |
| subj05 | **NeurIPS (Ours)** | 0.239 | 0.368 | 90.6% | 95.6% | 93.9% | 94.1% | 0.645 | 0.395 |
| subj07 | Yu et al. (2025) | 0.157 | 0.298 | 78.0% | 88.3% | 83.2% | 86.7% | 0.746 | 0.409 |
| subj07 | SIM (Dahan et al., 2025) | 0.113 | 0.255 | 79.3% | 88.8% | 85.0% | 88.4% | 0.749 | 0.461 |
| subj07 | **NeurIPS (Ours)** | 0.226 | 0.364 | 89.3% | 94.4% | 90.6% | 92.6% | 0.679 | 0.415 |

*Table 5.* Quantitative results for each subject on NSD `test` (8 subjects training).

| Subject | Method | Low-Level | | | | High-Level | | | |
|---------|--------|-----------|---|---|---|------------|---|---|---|
| | | PixCor↑ | SSIM↑ | Alex(2)↑ | Alex(5)↑ | Incep↑ | CLIP↑ | Eff↓ | SwAV↓ |
| subj01 | Yu et al. (2025) | 0.094 | 0.257 | 74.0% | 83.5% | 80.2% | 83.0% | 0.834 | 0.519 |
| subj01 | SIM (Dahan et al., 2025) | 0.123 | 0.264 | 81.3% | 90.0% | 85.0% | 87.9% | 0.761 | 0.467 |
| subj01 | **NeurIPS (Ours)** | 0.255 | 0.371 | 91.4% | 95.4% | 91.7% | 93.3% | 0.667 | 0.402 |
| subj02 | Yu et al. (2025) | 0.095 | 0.261 | 75.6% | 86.5% | 80.8% | 84.0% | 0.836 | 0.518 |
| subj02 | SIM (Dahan et al., 2025) | 0.109 | 0.258 | 79.2% | 88.8% | 83.8% | 86.8% | 0.771 | 0.472 |
| subj02 | **NeurIPS (Ours)** | 0.242 | 0.370 | 90.0% | 94.3% | 90.5% | 91.3% | 0.686 | 0.417 |
| subj03 | Yu et al. (2025) | 0.083 | 0.257 | 71.1% | 80.2% | 74.9% | 77.7% | 0.882 | 0.555 |
| subj03 | SIM (Dahan et al., 2025) | 0.092 | 0.259 | 74.5% | 81.7% | 74.7% | 77.3% | 0.852 | 0.535 |
| subj03 | **NeurIPS (Ours)** | 0.204 | 0.362 | 84.2% | 90.1% | 83.7% | 86.4% | 0.769 | 0.477 |
| subj04 | Yu et al. (2025) | 0.072 | 0.256 | 71.8% | 80.5% | 75.6% | 76.0% | 0.856 | 0.548 |
| subj04 | SIM (Dahan et al., 2025) | 0.080 | 0.256 | 72.8% | 80.5% | 76.1% | 79.3% | 0.841 | 0.533 |
| subj04 | **NeurIPS (Ours)** | 0.200 | 0.360 | 84.2% | 90.1% | 83.7% | 86.4% | 0.769 | 0.477 |
| subj05 | Yu et al. (2025) | 0.095 | 0.257 | 76.6% | 88.7% | 85.0% | 87.7% | 0.792 | 0.486 |
| subj05 | SIM (Dahan et al., 2025) | 0.115 | 0.262 | 80.4% | 89.5% | 86.5% | 88.7% | 0.744 | 0.456 |
| subj05 | **NeurIPS (Ours)** | 0.228 | 0.366 | 89.4% | 95.6% | 93.6% | 94.1% | 0.650 | 0.396 |
| subj06 | Yu et al. (2025) | 0.082 | 0.254 | 71.6% | 86.5% | 78.3% | 75.9% | 0.857 | 0.545 |
| subj06 | SIM (Dahan et al., 2025) | 0.089 | 0.250 | 72.6% | 81.0% | 76.1% | 79.0% | 0.843 | 0.533 |
| subj06 | **NeurIPS (Ours)** | 0.201 | 0.360 | 84.5% | 89.4% | 84.6% | 86.4% | 0.763 | 0.473 |
| subj07 | Yu et al. (2025) | 0.095 | 0.261 | 75.6% | 86.5% | 80.8% | 84.0% | 0.836 | 0.518 |
| subj07 | SIM (Dahan et al., 2025) | 0.105 | 0.251 | 77.5% | 87.1% | 82.3% | 86.1% | 0.776 | 0.481 |
| subj07 | **NeurIPS (Ours)** | 0.216 | 0.363 | 88.3% | 93.7% | 90.0% | 91.7% | 0.688 | 0.419 |
| subj08 | Yu et al. (2025) | 0.070 | 0.255 | 70.0% | 77.1% | 68.4% | 71.6% | 0.886 | 0.551 |
| subj08 | SIM (Dahan et al., 2025) | 0.074 | 0.247 | 70.6% | 76.6% | 69.1% | 73.2% | 0.887 | 0.571 |
| subj08 | **NeurIPS (Ours)** | 0.194 | 0.361 | 81.5% | 86.8% | 78.4% | 82.2% | 0.814 | 0.504 |

*Table 6.* Brain captioning results for each subject on NSD `test` (4 subjects training). Our model surpasses the baselines, highlighting the applicability of our method across various tasks.

| Subject | Method | BLEU1 ↑ | BLEU2 ↑ | BLEU3 ↑ | BLEU4 ↑ | METEOR ↑ | ROUGE ↑ | CIDEr ↑ | SPICE ↑ | CLIP-S ↑ | RefCLIP-S ↑ |
|---|---|---|---|---|---|---|---|---|---|---|---|
| subj01 | Yu et al. (2025) | 49.66 | 31.58 | 19.57 | 12.23 | 16.34 | 36.02 | 40.07 | 9.83 | 59.35 | 65.67 |
| subj01 | SIM (Dahan et al., 2025) | 50.08 | 31.78 | 20.06 | 12.66 | 16.66 | 36.83 | 41.01 | 9.85 | 60.71 | 67.26 |
| subj01 | **NeurIPS (Ours)** | 54.57 | 36.13 | 23.97 | 15.89 | 18.94 | 39.83 | 55.40 | 11.87 | 64.67 | 71.00 |
| subj02 | Yu et al. (2025) | 49.37 | 30.92 | 19.22 | 12.17 | 15.99 | 36.07 | 39.48 | 9.38 | 59.08 | 65.59 |
| subj02 | SIM (Dahan et al., 2025) | 49.55 | 31.05 | 19.48 | 12.50 | 16.46 | 36.33 | 40.08 | 9.86 | 60.27 | 66.85 |
| subj02 | **NeurIPS (Ours)** | 52.58 | 34.31 | 22.24 | 14.48 | 18.05 | 38.53 | 50.53 | 11.47 | 63.19 | 69.47 |
| subj05 | Yu et al. (2025) | 49.70 | 31.60 | 19.97 | 12.82 | 16.77 | 36.61 | 42.63 | 10.83 | 60.17 | 66.49 |
| subj05 | SIM (Dahan et al., 2025) | 50.92 | 32.70 | 20.91 | 13.61 | 17.10 | 37.45 | 44.34 | 10.26 | 61.32 | 67.92 |
| subj05 | **NeurIPS (Ours)** | 55.36 | 36.66 | 23.94 | 15.73 | 19.59 | 40.41 | 57.91 | 13.05 | 65.85 | 72.05 |
| subj07 | Yu et al. (2025) | 48.61 | 30.09 | 18.43 | 11.47 | 15.97 | 35.87 | 37.38 | 9.33 | 58.05 | 64.45 |
| subj07 | SIM (Dahan et al., 2025) | 49.77 | 31.43 | 19.68 | 12.33 | 16.30 | 36.21 | 39.15 | 9.39 | 59.67 | 66.48 |
| subj07 | **NeurIPS (Ours)** | 53.36 | 35.11 | 22.88 | 15.11 | 18.36 | 39.40 | 51.06 | 11.82 | 63.25 | 69.72 |
| Average | Yu et al. (2025) | 49.33 | 31.05 | 19.30 | 12.17 | 16.27 | 36.14 | 39.89 | 9.84 | 59.16 | 65.55 |
| Average | SIM (Dahan et al., 2025) | 50.08 | 31.74 | 20.03 | 12.77 | 16.63 | 36.70 | 41.14 | 9.84 | 60.49 | 67.13 |
| Average | **NeurIPS (Ours)** | **53.96** | **35.55** | **23.26** | **15.30** | **18.73** | **39.54** | **53.72** | **12.05** | **64.24** | **70.56** |

*Table 7.* Quantitative results for brain retrieval on the NSD `test`. The brain retrieval pipeline follows MindEye (Scotti et al., 2023). For each retrieval task, top-1 accuracy (`Acc@1`) and top-5 accuracy (`Acc@5`) are reported. The 95% confidence interval (CI) is also reported (mean±CI). The results of the retrieval task align with those of image reconstruction, demonstrating the powerful capability of our designed semantic decoder.

| # | Method | Brain-to-Image (%) | | Brain-to-Text (%) | | Image-to-Brain (%) | | Text-to-Brain (%) | |
|---|---|---|---|---|---|---|---|---|---|
| | | Acc@1 ↑ | Acc@5 ↑ | Acc@1 ↑ | Acc@5 ↑ | Acc@1 ↑ | Acc@5 ↑ | Acc@1 ↑ | Acc@5 ↑ |
| | Yu et al. (2025) | $68.9_{\pm 2.9}$ | $89.6_{\pm 2.4}$ | $49.0_{\pm 2.7}$ | $71.8_{\pm 2.5}$ | $61.2_{\pm 2.7}$ | $85.7_{\pm 2.3}$ | $57.2_{\pm 3.1}$ | $79.6_{\pm 2.8}$ |
| | SIM (Dahan et al., 2025) | $82.9_{\pm 2.5}$ | $93.5_{\pm 1.8}$ | $66.4_{\pm 2.5}$ | $78.8_{\pm 2.2}$ | $80.5_{\pm 2.4}$ | $92.1_{\pm 1.7}$ | $73.6_{\pm 2.6}$ | $88.6_{\pm 2.1}$ |
| 1 | w/o global token | $82.5_{\pm 1.2}$ | $96.2_{\pm 0.6}$ | $69.5_{\pm 1.3}$ | $87.2_{\pm 1.1}$ | $74.2_{\pm 1.4}$ | $91.9_{\pm 1.0}$ | $70.0_{\pm 1.4}$ | $88.6_{\pm 1.1}$ |
| 2 | subject ID gating | $89.8_{\pm 1.0}$ | $99.3_{\pm 0.3}$ | $77.8_{\pm 1.2}$ | $97.5_{\pm 0.5}$ | $81.5_{\pm 1.2}$ | $99.0_{\pm 0.3}$ | $76.4_{\pm 1.2}$ | $98.0_{\pm 0.4}$ |
| 5 | Yu-style structure fusion | $88.2_{\pm 1.0}$ | $98.1_{\pm 0.5}$ | $76.6_{\pm 1.3}$ | $95.4_{\pm 0.6}$ | $79.5_{\pm 1.2}$ | $98.6_{\pm 0.4}$ | $73.3_{\pm 1.2}$ | $96.7_{\pm 0.5}$ |
| 6 | full brain | $83.9_{\pm 1.2}$ | $97.9_{\pm 0.5}$ | $63.5_{\pm 1.4}$ | $86.4_{\pm 1.1}$ | $78.1_{\pm 1.3}$ | $96.4_{\pm 0.6}$ | $74.2_{\pm 1.3}$ | $94.2_{\pm 0.7}$ |
| 7 | functional features gating | $88.9_{\pm 1.0}$ | $99.5_{\pm 0.2}$ | $77.7_{\pm 1.2}$ | $97.7_{\pm 0.5}$ | $81.3_{\pm 1.2}$ | $98.7_{\pm 0.4}$ | $76.6_{\pm 1.2}$ | $98.1_{\pm 0.4}$ |
| 8 | w/o left brain tokens | $84.7_{\pm 1.1}$ | $96.8_{\pm 0.6}$ | $73.1_{\pm 1.4}$ | $91.1_{\pm 0.9}$ | $77.2_{\pm 1.4}$ | $94.8_{\pm 0.7}$ | $72.5_{\pm 1.2}$ | $92.2_{\pm 0.8}$ |
| 9 | w/o right brain tokens | $85.8_{\pm 1.1}$ | $96.8_{\pm 0.6}$ | $73.3_{\pm 1.4}$ | $91.0_{\pm 0.9}$ | $76.7_{\pm 1.3}$ | $94.9_{\pm 0.7}$ | $72.5_{\pm 1.3}$ | $91.8_{\pm 0.9}$ |
| 10 | shuffle spherical position | $86.7_{\pm 1.1}$ | $97.4_{\pm 0.6}$ | $74.3_{\pm 1.3}$ | $91.2_{\pm 0.8}$ | $76.5_{\pm 1.4}$ | $95.0_{\pm 0.7}$ | $71.1_{\pm 1.4}$ | $92.2_{\pm 0.8}$ |
| 11 | convolution receptive field = 1 | $88.1_{\pm 1.1}$ | $97.8_{\pm 0.5}$ | $74.6_{\pm 1.3}$ | $92.0_{\pm 0.8}$ | $78.1_{\pm 1.3}$ | $95.4_{\pm 0.7}$ | $72.5_{\pm 1.4}$ | $92.4_{\pm 0.8}$ |
| 12 | **NeurIPS (Ours) (semantic decoding only)** | $91.1_{\pm 0.9}$ | $99.7_{\pm 0.2}$ | $78.9_{\pm 1.3}$ | $97.8_{\pm 0.4}$ | $82.2_{\pm 1.2}$ | $99.1_{\pm 0.3}$ | $77.0_{\pm 1.1}$ | $98.4_{\pm 0.4}$ |

*Table 8.* **Few-shot adaptation under severe data limitations.** After multi-subject pretraining, we fine-tune the model on the held-out subject (Subj01) using very limited data budgets for 10 epochs. Even with only 5% of the target subject's data, NeurIPS retains strong high-level semantic decoding capabilities.

| Data Fraction | Fine-tuning Epochs | Alex(5) ↑ | CLIP ↑ |
|---|---|---|---|
| 5% | 10 | 86.0% | 80.0% |
| 20% | 10 | 93.3% | 91.2% |

*Table 9.* **Anatomy similarity vs. SG-MoE routing similarity.** We quantify the pairwise anatomical similarity (based on thickness, curvature, and sulcal depth) and the corresponding routing decision similarity between held-out subjects. Subject pairs with higher anatomical similarity tend to exhibit more similar expert routing patterns.

| Subject Pair | Anatomy Similarity | SG-MoE Routing Similarity |
|---|---|---|
| subj01, subj02 | 0.1555 | 0.7627 |
| subj01, subj05 | 0.1722 | 0.7654 |
| subj01, subj07 | 0.1280 | 0.7543 |

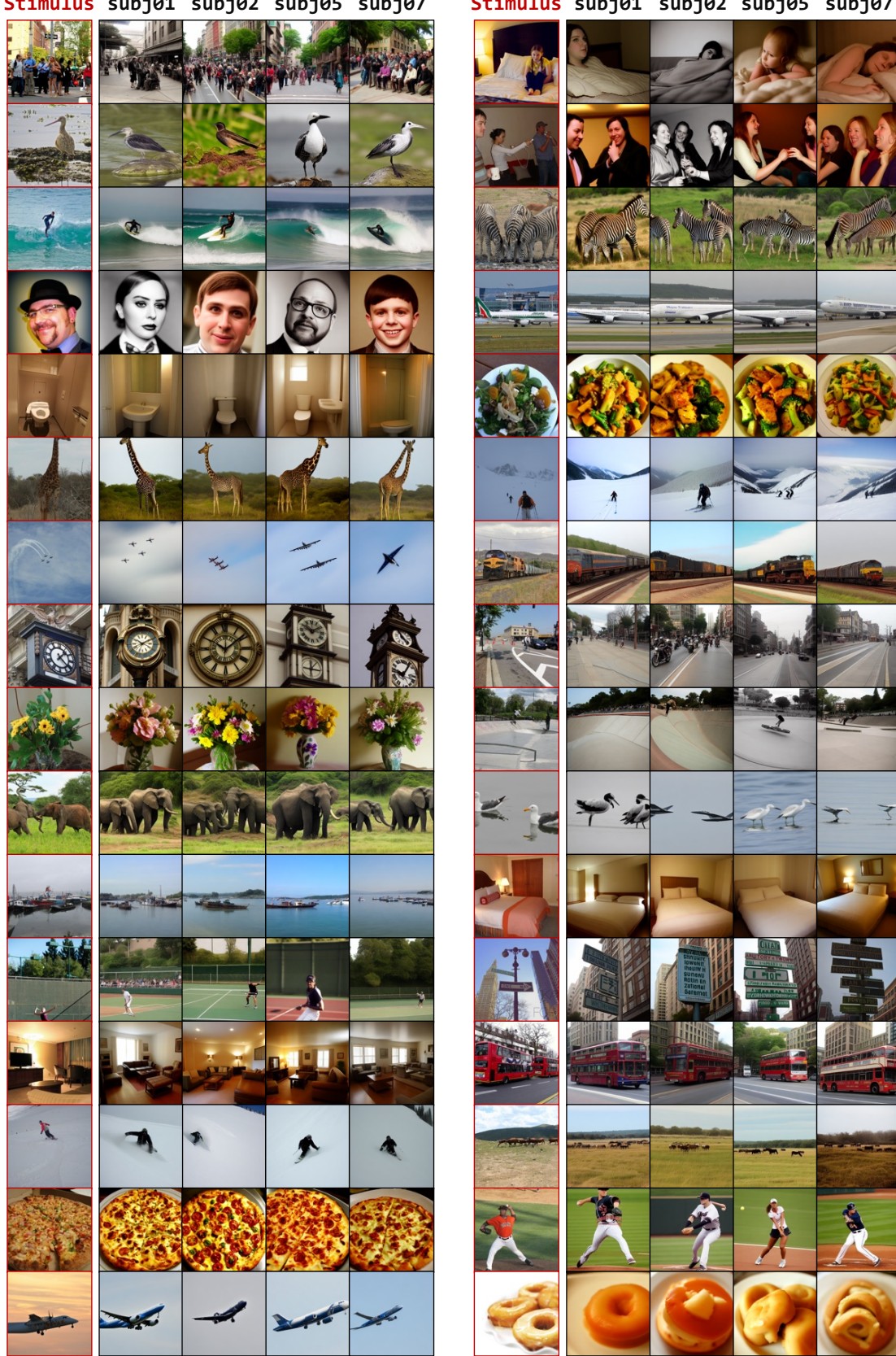

*Figure 13.* More visual reconstruction results for different subjects on NSD `test`.

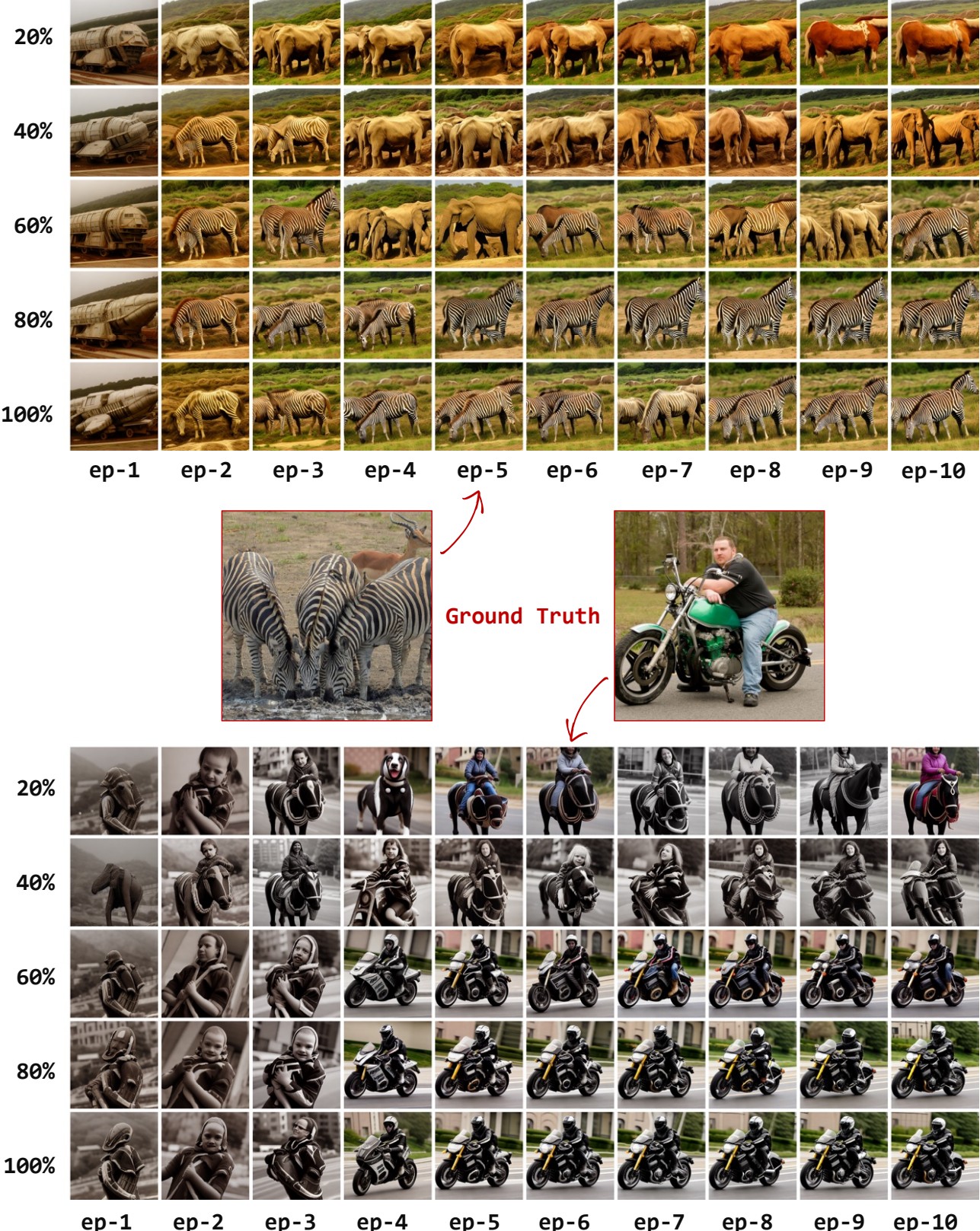

*Figure 14.* More visual reconstruction results for fine-tuning on NSD `test`. We pretrain the NeurIPS on three subjects (subj02, subj05, subj07) and fine-tune it on a completely new subject (subj01) under limited data ($y$-axis) and time constraints ($x$-axis). Overall, NeurIPS is able to achieve satisfactory reconstruction results under stringent constraints, demonstrating its strong cross-subject generalization ability and adaptability to new subjects.

