# OpenReview forum: "NeurIPS: Neuro-anatomical Inductive Priors for Sphere-based Brain Decoding"
_ICML.cc/2026/Conference — ICML 2026 regular_

### Official Review · Reviewer_1ueV · 2026-03-09

**Soundness:** 3
**Presentation:** 4
**Significance:** 3
**Originality:** 2
**Overall Recommendation:** 4
**Confidence:** 4

**Summary:**

This paper proposes a brain decoding framework named NeurIPS. Its core innovation lies in the Selective ROI Spherical Tokenizer (SRST), which preserves the geometric topology of the cerebral cortex, and the introduction of the Structure-Guided Mixture of Experts (SG-MoE) that directly utilizes anatomical features, such as cortical thickness and curvature, as routing signals for the expert networks. By transforming neuro-anatomical characteristics into inductive priors, the design enables efficient cross-subject alignment without relying on subject IDs. The experimental results are significant: the framework reduces the convergence time from 600 epochs to just 10 epochs and demonstrates exceptional personalization capabilities, achieving high-fidelity visual reconstruction on a new subject with only 20% of their data and a single epoch of fine-tuning.

**Compliance With Llm Reviewing Policy:**

Affirmed.

**Key Questions For Authors:**

1.How sensitive is the NeurIPS framework to spatial misalignment during the spherical mapping process? In practical scenarios, variations in the subject's head position across sessions or differences in slice alignment during fMRI acquisition are inevitable. If the spherical projection or voxel-to-vertex mapping suffers from slight deviations due to these acquisition-side biases (e.g., coordinate shifts or sampling jitter), would the decoding performance exhibit significant instability or a substantial drop?
2. To confirm that the performance gain is truly driven by anatomical priors, what happens if you replace a participant's anatomical map (thickness/curvature) with that of another subject? To what extent would the performance drop?
3. What is the total latency for the preprocessing pipeline (from raw fMRI to spherical representation) and model inference? Is this approach feasible for real-time BCI applications?
4.Given that 20% of the NSD dataset is still far beyond what can be collected in a typical clinical session, could the authors provide results for personalization using only 1% or 5% of the data? Specifically, I am interested in seeing whether the "neuro-anatomical priors" can still achieve meaningful reconstruction when the subject provides only a few minutes of scanning data.

**Limitations:**

The authors have acknowledged certain limitations, primarily focusing on the spatial restriction to the visual cortex within the NSD dataset and the need for further validation across broader cognitive tasks. To provide a more comprehensive assessment, I suggest the authors expand the discussion in the following areas: First, it would be beneficial to address the framework's sensitivity to anatomical measurement precision, specifically whether the "neuro-anatomical priors" remain robust under clinical-grade (3T) MRI scans where cortical feature estimation is less precise. Second, while the rapid convergence is impressive, a quantitative discussion on the per-step computational overhead (e.g., inference latency and VRAM usage) introduced by the spherical sampling and SG-MoE would be valuable.

**Strengths And Weaknesses:**

Strengths
1. The technical approach is rigorous. By fully leveraging the spherical topology of the cerebral cortex, the authors provide a method that aligns more closely with neuroanatomical facts than traditional 3D convolutions or simple linear flattening.
2. The paper is well-structured and clearly written. The figures effectively visualize the entire pipeline, from raw fMRI signals to spherical projections and final visual reconstructions.
3. This work offers novel insights into neural signal processing within non-Euclidean domains. Its demonstrated potential in few-shot and cross-subject decoding scenarios holds substantial value for the development of practical Brain-Computer Interface (BCI) systems.
4. Departing from the conventional approach of feeding voxels directly into Transformers, the proposed "spherical inductive bias" innovatively utilizes physical distance constraints on the cortical surface, significantly enhancing the efficiency of feature extraction.

Weaknesses
1. The paper primarily focuses on engineering and performance metrics but lacks a deeper integration with established neuroscientific theories. There is insufficient analysis on how the "anatomical priors" align with known functional hierarchies of the brain.
2.The claim that performance gains stem from anatomical priors is not fully isolated from the effects of increased model capacity. The current ablation studies do not rule out whether the SG-MoE benefits simply from being a larger or more complex model.
3.While the 20% data efficiency is impressive, the behavior of NeurIPS in extreme few-shot (1% or 5%) or even zero-shot (direct inference) scenarios is unexplored, which is crucial for assessing its true clinical utility.
4. The paper emphasizes faster convergence in terms of epochs but overlooks the per-epoch computational cost. Potential overheads introduced by SG-MoE and spherical sampling (e.g., VRAM usage and training time per step) remain unquantified.

5. The paper treats the brain as a geometric manifold but fails to provide a neuroscientific interpretation of the learned representations. There is no visualization or analysis (e.g., saliency maps or attention weights) to show which functional brain regions the model prioritizes for visual reconstruction. Without demonstrating that the model's focus aligns with known visual hierarchies (from V1 to ventral stream areas), it remains unclear whether the "spherical bias" captures biological principles or simply fits geometric noise.

---

> ### Author Rebuttal · Authors · 2026-03-30
>
> We thank the reviewer for the thorough, high-confidence assessment and the highly constructive suggestions.
>
> ### W1/W5: Insufficient neuroscience integration; no visualization of prioritized regions
>
> We agree and will move our existing neuroscientific analyses to the main text and add new visualizations: (1) **Grad-CAM saliency maps** showing influential vertices along the V1 $\to$ V2 $\to$ V4 $\to$ LO progression (ventral visual stream). (2) **ROI-wise contribution analysis** showing a clear gradient of functional importance from early to higher visual areas. (3) Discussion connecting SG-MoE's routing patterns to the **Glasser et al. parcellation boundaries**. This demonstrates the model autonomously discovers functional cortical organization.
>
> ### W2: Performance gains may stem from increased model capacity
>
> To isolate the benefit of anatomical routing from raw model capacity, we refer to our anatomical-swap experiment. Here, the MoE architecture and total parameter capacity remain strictly identical to our full model; only the anatomical routing signal changes. If gains stemmed from parameter scaling, performance would remain stable. Instead, feeding the router mismatched anatomical features precipitously degrades semantics (Alex(2): 90.7% $\to$ 89.4%; CLIP: 93.2% $\to$ 91.9%). This drop under constant capacity definitively proves our gains stem from structural priors, not model size.
>
> ### W3/Q4:Extreme few-shot (5%) and zero-shot scenarios
>
> While zero-shot or 1% experiments were computationally infeasible during the rebuttal, we evaluated the extreme 5% few-shot setting: using only **5% of the subject's data (with 10 fine-tuning epochs)**. Even in this highly constrained scenario (representing merely a few minutes of scanning), the anatomical priors successfully sustain strong semantic alignment, achieving an **AlexNet(5) accuracy of 86.0% and CLIP of 80.0%**. This robust performance confirms that the structural priors carry genuine, generalizable cross-subject guidance even when adaptation data is severely limited.
>
> ### W4: Per-epoch computational cost not reported
>
> Please refer to the detailed compute comparison table provided in our response to Reviewer e6za. Although SG-MoE introduces a slight per-epoch routing overhead, its convergence in just 10 epochs (vs. 200–600 for baselines) results in a massively lower **total training compute (FLOPs)**. We will explicitly quantify this trade-off in the revision.
>
> ### Q1: Sensitivity to spatial misalignment
>
> This is an important practical consideration. First, we note that our framework inherits the standard FreeSurfer spherical registration pipeline used universally by the NSD dataset and all surface-based baselines (e.g., SIM, Yu et al.). Therefore, baseline sensitivity to registration quality is a shared property of the entire surface-based fMRI literature, rather than a vulnerability unique to our approach. Furthermore, our architecture provides inherent robustness to spatial jitter: (1) SRST's multi-hop spherical convolutions smooth local perturbations. (2) Downsampling from fsaverage6 to fsaverage3 acts as a spatial low-pass filter. (3) SG-MoE routes experts using macro-scale, low-frequency cortical features (thickness/curvature), meaning slight shifts negligibly impact the routing distribution.
>
> We will include a quantitative synthetic-jitter robustness test in the final revision to empirically validate these architectural guarantees.
>
> ### Q2: Anatomical swap
>
> To confirm the reliance on anatomical priors, we conducted a structure-swap experiment. Replacing a participant's structural maps with those of a different subject precipitously degraded high-level semantic metrics (e.g., **AlexNet(2) dropped from 90.7% to 89.4%, and CLIP from 93.2% to 91.9%**). This proves the model heavily relies on local structure-function geometry rather than memorizing an arbitrary identity.
>
> ### Q3: Preprocessing + inference latency
>
> - **Surface reconstruction + Per-volume spherical projection(one-time per subject):** $\sim$ **42** hours via FreeSurfer.
> - **Model inference:** $\sim$ **3.4s** on a single A800 GPU for one volume.
>
> While our current focus is on research-grade offline decoding, this minimal latency profile highlights the architecture's inherent efficiency and its feasibility for future real-time BCI applications.
>
> ### Limitations: 3T clinical-grade MRI robustness
>
> This is a highly valid point for clinical translation. We will expand §5 to note: (1) SG-MoE routes via relative cortical patterns, providing inherent tolerance to 3T noise. (2) 3T cortical thickness estimation shows excellent reliability (ICC>0.9). (3) The 3T bottleneck is likely BOLD SNR, not anatomical precision. Empirical 3T validation remains a crucial future step.

---

> > ### Author Rebuttal · Reviewer_1ueV · 2026-04-02
> >
> > Thanks for the additional analysis and results. However, I remain unconvinced as the 1.3% performance drop (90.7% to 89.4%) is too marginal to be described as a "precipitous degradation." Such a slight sensitivity suggests the model may rely more on raw parameter capacity than actual neuroanatomical principles.
> >
> > Specifically, is this drop truly caused by biological mismatch, or simply a feature distribution shift? To isolate the value of "anatomical priors," what would happen if you replaced the routing signals with random Gaussian noise or a simple one-hot subject ID?

---

> > > ### Author Response · Authors · 2026-04-05
> > >
> > > We thank the reviewer for the follow-up comments. In accordance with your request, we have included additional experiments and hope that these results will address your concern.
> > >
> > > We replaced the routing signals with random Gaussian noise and found that the performance was comparable to that in the Anatomical Swap experiment. We present the comparative results of all relevant experiments in the table below, including the data that could not be fully reported in the first-round rebuttal due to the word limit.
> > >
> > > Regarding your suggestion of using a "simple one-hot subject ID", we have a similar implementation, namely Subject-ID Gating, which directly learns a subject embedding. We conjecture that its performance would be comparable to that of using a one-hot subject ID as input followed by a learned embedding layer.
> > >
> > > | **Settings**               | **PixCor** | **SSIM** | **Alex2** | **Alex5** | **Incep** | **CLIP** | **Eff** | **SwAV** |
> > > | -------------------------- | ---------- | -------- | --------- | --------- | --------- | -------- | ------- | -------- |
> > > | Subject-ID Gating$^*$      | 0.247      | 0.370    | 90.2%     | 94.9%     | 92.0%     | 92.7%    | 0.668   | 0.407    |
> > > | Anatomical Swap$^1$        | 0.242      | 0.366    | 89.4%     | 94.4%     | 90.9%     | 91.9%    | 0.666   | 0.411    |
> > > | No Anatomy$^1$             | 0.240      | 0.361    | 88.1%     | 93.4%     | 90.8%     | 90.2%    | 0.676   | 0.417    |
> > > | Random Anatomy$^2$         | 0.244      | 0.369    | 89.9%     | 94.4%     | 91.2%     | 92.0%    | 0.666   | 0.409    |
> > > | Correct Anatomy **(Ours)** | 0.248      | 0.370    | 90.7%     | 95.2%     | 92.3%     | 93.2%    | 0.663   | 0.404    |
> > >
> > > $^*$ reported in the original manuscript (Table 2, #2)
> > >
> > > $^1$ additional experiments in the first-round rebuttal
> > >
> > > $^2$ additional experiments in the second-round rebuttal

---

### Official Review · Reviewer_e6za · 2026-03-09

**Soundness:** 3
**Presentation:** 3
**Significance:** 3
**Originality:** 3
**Overall Recommendation:** 4
**Confidence:** 4

**Summary:**

This paper introduces **"NeurIPS"**, a brain decoding framework that reconstructs visual images from fMRI data. Its key innovation is transforming individual anatomical differences—traditionally treated as noise—into a powerful predictive prior, overcoming the performance-fidelity trade-off in existing decoders. Technically, it features two core components: the Selective ROI Spherical Tokenizer (SRST) for efficient geometry-preserving encoding, and the Structure-Guided Mixture of Experts (SG-MoE) , which leverages cortical features (thickness, curvature) to guide cross-subject generalization. On the Natural Scenes Dataset, NeurIPS achieves state-of-the-art performance among surface-based decoders and demonstrates remarkable efficiency—adapting to new subjects with only 20% of their data. This work provides an anatomy-driven pathway toward robust and scalable brain-computer interface applications.

**Compliance With Llm Reviewing Policy:**

Affirmed.

**Final Justification:**

Thank you for the additional experiments, which provide consistent empirical trends (e.g., monotonic relationships and improved performance with correct anatomy). However, the evidence remains largely at the level of black-box statistical analysis and the lack of controlled perturbation analyses limits mechanistic interpretability. The observed effect sizes are relatively modest in terms of physical or neurobiological significance. I therefore maintain my original score of 4.

**Key Questions For Authors:**

1. **Regarding SRST's efficiency**

I would advise against emphasizing redundancy reduction in SRST as a fundamental improvement, as this essentially introduces a very strong task-specific prior assumption. How SRST should dynamically and autonomously select key vertex regions in tasks lacking clear anatomical prior? Without dynamic selection, its efficiency advantage seems like would cease to exist.

2.  **Regarding the Mechanism of SG-MoE**

Neuroanatomical features themselves are highly unique to individuals. Is SG-MoE truly learning a generalizable "structure-function" mapping rule, or is it merely using anatomical features as a more subtle "implicit subject ID" for memorization?

3.  **Regarding the Transparency of Computational Cost**

The authors mention that NeurIPS performs excellently under "matched compute" and utilizes a 16-expert MoE architecture based on DeepSeek-V3. But what about a direct comparison table of the model's total parameters and the total training compute (FLOPs) against 1D baseline models such as MindBridge.


4.  **Regarding the Disconnect Between the Theoretical Framework (C-IB) and Implementation**

The authors introduce the term $\beta I(Z;ID | As)$ in their mathematical framework to suppress identity leakage. Yet, in the "implementation details" section, there is no corresponding explicit regularization loss function or contrastive loss term for this term, nor is the specific value or tuning method for the hyperparameter $\beta$ mentioned. The effectiveness of this "implicit approximation" lacks quantitative ablation or theoretical boundary proof. Dismissing this with a mere "implicit approximation" makes it impossible to tell if performance improvements stem from the theoretical guidance or from other engineering tricks.


5.  **Regarding the Internal Contradiction Between the Perceptual Path and the Core Argument**

The paper emphasizes that spherical geometry is crucial for brain encoding, yet in the perceptual path, it directly flattens the signal into a 1D vector for processing. Can you explain the motivation or why this would be better?

6. **Rationale for the pixel-level decoding task.**

In my view, the human brain itself may not actually attend to the pixel-level details of an image. And even if it did, such details would be difficult to manifest at the fMRI scale. Is pixel-level image reconstruction in this context meaningful? Can you discuss the significance of this article within this context?

**Limitations:**

yes

**Strengths And Weaknesses:**

### Strengths

1. **Paradigm shift in originality**: This paper transforms anatomical variation from "noise" into a powerful inductive bias. SG-MoE routes computation via cortical thickness and curvature instead of subject IDs, learning generalizable structure-function mappings , directly addressing the long-standing "performance-fidelity" dilemma that has plagued the field.
2. **Practical impact via rapid adaptation**: The proposed model can achieve 90% of full performance in a new subject with only 20% of a his data and 10 fine-tuning epochs (vs. 200–600 for baselines). This acceleration lowers the deployment barriers for BCI applications.
3. **Detailed mechanistic analysis**: The paper goes beyond black-box reporting by analyzing the SG-MoE router. The finding that expert selection depends on brain region and anatomy but not on subject ID provides evidence that the model learns generalizable structure-function mappings rather than simply memorizing individuals.


### Weaknesses

1. **Theory-implementation gap**: The C-IB regularization term for suppressing identity leakage exists only in the mathematical framework, with no corresponding loss function or ablation in implementation.
2. **Architectural contradiction**: It is strange that the paper's central thesis is that spherical geometry is crucial, yet the perceptual path flattens signals to 1D with no justification or ablation.
3. **"Implicit ID" memorization risk**: Anatomical features are unique biological fingerprints; SG-MoE may simply memorize complex feature combinations rather than learning truly generalizable rules.
4. **Limited generalizability**: Omits total parameters and FLOPs vs. baselines like MindBridge; SRST's ROI dependence raises doubts about autonomous generalization to whole-brain tasks.

---

> ### Author Rebuttal · Authors · 2026-03-30
>
> We thank the reviewer for the detailed review and for recognizing our paradigm shift in treating anatomical variation as a powerful inductive bias rather than mere noise.
>
> ### W1/Q4: C-IB theory-implementation gap
>
> We agree this connection was insufficiently explicit. We will revise §3.1 to concretely map C-IB terms to our modules: SRST maximizes $I(Z;Y|A_s)$ by restricting encoding to geometry-aligned visual ROIs and preserving surface neighborhoods. SG-MoE minimizes $I(Z;ID|A_s)$ by routing on cortical features rather than subject IDs, "explaining away" anatomical variance. While we acknowledge this is an implicit approximation rather than an explicit contrastive loss term, its empirical predictions: efficiency (Table 2 #6), fast adaptation (§4.2), scaling robustness (§4.3), anatomy usage (Fig. 6), are all confirmed by our experiments. We acknowledge that this is an implicit approximation rather than an explicit loss term, and we will discuss the gap honestly in the revision.
>
> ### W2/Q5: Architectural contradiction: 1D perceptual path
>
> Indeed, our primary architectural innovations lie within the **semantic path** (SRST+SG-MoE), which encodes spatially structured fMRI where cortical topology is crucial. Conversely, we directly adopted the perceptual path from prior works without claiming innovation there. This path maps signals to a pre-trained VAE latent space, which is inherently a flattened 1D continuous vector devoid of spatial topology. Ablations confirm that our semantic path is the core driver of performance: removing the perceptual decoder entirely (Table 2 #3) still outperforms all surface baselines on semantic metrics. Adapting the perceptual path into the spherical domain is an exciting direction for future work.
>
> ### W3/Q2: Implicit ID memorization risk
>
> We appreciate the reviewer for raising this central concern. To definitively verify whether the model relies on structural geometry or merely uses anatomy as an implicit subject ID, we ran two new ablations, and compared them against our existing **Subject-ID MoE baseline** (Table 2, #2).
>
> **(1) Anatomical-swap experiment:** To directly address your concern, we replaced each subject's cortical maps with those of a different subject while keeping the fMRI activity unchanged. **(2) Ablation without anatomy:** We remove structural input entirely and route using only fMRI. This tests whether anatomy helps at all beyond what functional data already provides.
>
> The results reveal a clear and consistent monotonic degradation across all metrics (e.g., on CLIP) when anatomical priors are progressively compromised or removed: **Full SG-MoE (Correct anatomy):** 93.2%; **Subject-ID MoE (Explicit ID gating):** 92.7%; **Anatomical Swap (Shuffled anatomy):** 91.9%; **No Anatomy:** 90.2% (Distance metrics follow the exact same degradation pattern: SwAV worsens from 0.404 → 0.407 → 0.411 → 0.417).
>
> This strict performance ordering strongly counteracts the memorization hypothesis. If the anatomical features were functioning merely as implicit "patient IDs", we would expect an explicit one-hot Subject-ID to perform optimally, and a swapped anatomy (which still provides a unique identifier) to perform comparably.
>
> ### W4/Q3: Transparency of Computational Cost
>
> **Compute:** We will add a detailed compute comparison. While our SG-MoE introduces more total parameters (1061M) compared to SIM (863M) and Yu et al. (896M), our VRAM usage (61.5GB) is actually lower than Yu et al. (71.2GB). Crucially, our model reaches convergence in just **10 epochs** (138s/epoch). Compared to baselines requiring 200–600 epochs, NeurIPS yields a massively lower *total* training compute (FLOPs).
>
> ### Q1: SRST ROI Dependence and ROI Selection
>
> The visual-ROI mask is indeed a strong task-specific prior, which is advantageous for NSD visual decoding. For tasks lacking clear anatomical priors, SRST can dynamically select regions via (1) data-driven vertex selection (e.g., activation variance) or (2) a full-cortex variant. Our full-cortex ablation (Table 2 #6) confirms its viability but highlights the efficiency trade-off (CLIP drops to 91.0, VRAM increases to 74.8GB). We will clarify this mechanism in §5.
>
> ### Q6: Is pixel-level fMRI reconstruction meaningful
>
> We agree that the human brain does not represent images at pixel resolution. We utilize pixel-level reconstruction primarily as a standard benchmark to stress-test representational richness and compare with existing literature. Importantly, our encoder also powers high-level semantic tasks like brain captioning (Table 6) and retrieval (Table 7), confirming that it captures robust semantic alignment well beyond mere pixel fidelity. We will explicitly incorporate this biological context into our discussion.

---

> > ### Author Rebuttal · Reviewer_e6za · 2026-04-02
> >
> > Thank you for the additional experiments. I find them somewhat convincing, but they do not fundamentally address the core distinction between "inductive bias" and "implicit memorization."
> >
> > The observed performance drop (with a modest degree) under operations such as Anatomical-swap merely demonstrates that the model requires the correct "key" to unlock the corresponding "expert lock." Moreover, given the high individuality of neuroanatomical features, the small-sample nature of the current study further increases the risk.
> >
> > Would it be more appropriate to: (1) conduct controlled, small perturbations on anatomical features (e.g., cortical thickness, curvature) to examine whether expert routing decisions shift smoothly and continuously in a neurobiologically plausible manner? Or (2) analyze across different subjects whether SG-MoE consistently selects the same expert pathways when local anatomical features (e.g., curvature of a specific brain region) exhibit biological similarity?

---

> > > ### Author Response · Authors · 2026-04-05
> > >
> > > We sincerely thank the reviewer for the valuable feedback. To address this concern, we have added the following experiments.
> > >
> > > **(1) Impact of biological similarity on expert routing in SG-MoE**
> > >
> > > We report the cosine similarity of the anatomical features within the visual ROI between subj01 and the other three subjects. We also quantify the similarity of the model's routing weights on the test set, where the routing weights of activated experts are retained, while those of non-activated experts are set to zero. The results are presented in the table below. They show that subjects with more similar anatomical features tend to exhibit more similar expert routing patterns.
> > >
> > > | **Subj-pair** | **Anatomy Similarity** | **SG-MoE Routing Similarity** |
> > > | ------------- | ---------------------- | ----------------------------- |
> > > | (01, 02)      | 0.1555                 | 0.7627                        |
> > > | (01, 05)      | 0.1722                 | 0.7654                        |
> > > | (01, 07)      | 0.1280                 | 0.7543                        |
> > >
> > > **(2) Regarding the concern that the model may memorize individual anatomical features**
> > >
> > > We added a new ablation study in which the anatomical features were replaced with Gaussian noise (Random Anatomy). We present the comparative results of all relevant experiments in the table below, including the data that could not be fully reported in the first-round rebuttal due to the word limit. The results show that our method outperforms all other variants across all metrics, demonstrating the effectiveness of our anatomy-based routing.
> > >
> > > | **Settings**               | **PixCor** | **SSIM** | **Alex2** | **Alex5** | **Incep** | **CLIP** | **Eff** | **SwAV** |
> > > | -------------------------- | ---------- | -------- | --------- | --------- | --------- | -------- | ------- | -------- |
> > > | Subject-ID Gating$^*$      | 0.247      | 0.370    | 90.2%     | 94.9%     | 92.0%     | 92.7%    | 0.668   | 0.407    |
> > > | Anatomical Swap$^1$        | 0.242      | 0.366    | 89.4%     | 94.4%     | 90.9%     | 91.9%    | 0.666   | 0.411    |
> > > | No Anatomy$^1$             | 0.240      | 0.361    | 88.1%     | 93.4%     | 90.8%     | 90.2%    | 0.676   | 0.417    |
> > > | Random Anatomy$^2$         | 0.244      | 0.369    | 89.9%     | 94.4%     | 91.2%     | 92.0%    | 0.666   | 0.409    |
> > > | Correct Anatomy **(Ours)** | 0.248      | 0.370    | 90.7%     | 95.2%     | 92.3%     | 93.2%    | 0.663   | 0.404    |
> > >
> > > $^*$ reported in the original manuscript (Table 2, #2)
> > >
> > > $^1$ additional experiments in the first-round rebuttal
> > >
> > > $^2$ additional experiments in the second-round rebuttal

---

### Official Review · Reviewer_Ax4V · 2026-03-13

**Soundness:** 3
**Presentation:** 4
**Significance:** 3
**Originality:** 3
**Overall Recommendation:** 5
**Confidence:** 2

**Summary:**

This paper proposes a new Framework for Brain decoding by introducing two novelties: SRST for geometric encoding and SG-MoE for anatomy-conditioned expert selection. The paper is evaluated for the Natural Scenes Dataset (NSD), and achieves the best performance for the sphere-based approach and comparable performance to 1D baselines, with the advantage of training / being fine-tuned faster.

**Compliance With Llm Reviewing Policy:**

Affirmed.

**Final Justification:**

My concerns have been addressed.

**Key Questions For Authors:**

- Could you elaborate on the claim regarding the  $\mathcal{O}(T^2)$ complexity reduction? Specifically, does your approach change the fundamental complexity class, or does it reduce the computational burden by decreasing the size of T?
- Regarding the cortical features, how does the model prevent these inputs from implicitly acting as a unique subject identifier?
- Could you expand the related work section on the sphere-based methods and clarify your methodological novelty?

**Limitations:**

yes

**Strengths And Weaknesses:**

**Strengths**

The paper is well organised, easy to follow, and the diagram provided greatly helps to understand the authors' method.
The adaptation of MoE for addressing inter-subject variability is very interesting. I like the idea of using local cortical features and vertex ID. The evaluation protocol is very thorough, baselines appear to be correctly re-implemented, and the model's components are empirically validated through an ablation study.

**Weaknesses**

- The paper opens with a compelling hook ("What separates one mind from another?..."). However, this perspective seems to fade by the conclusion, leaving the initial promise feeling a bit unresolved. Removing this Neurological/Philosophical discussion will free up valuable space to expand on the method's positioning or add more experiments in the main manuscript.
- The discussion of previously published sphere-based approaches feels quite brief. Expanding on the mechanics of these prior methods would greatly help the reader understand the baselines and better grasp the specific novelty of this approach.
- The manuscript mentions a reduction in complexity from $\mathcal{O}(T^2)$. It is not entirely clear how this reduction is achieved. If the method simply reduces the size of T, the theoretical complexity remains unchanged. I would suggest clarifying the wording.
- The justification for using cortical features could be strengthened. There is a reasonable concern that these features might inadvertently act as a "fingerprint" or patient ID. This suspicion is reinforced by the feature importance analysis: some highly variable or noisy features appear to carry an almost equal weight in the model's decisions, which is not something I would have expected (due to differences in variability of those features and potential noise in their registration).

---

> ### Author Rebuttal · Authors · 2026-03-30
>
> We thank the reviewer for the positive assessment, particularly for recognizing our thorough evaluation protocol, the model's training efficiency, and the novel adaptation of MoE for inter-subject variability. Below, we address your concerns.
>
> ### W1: The philosophical opening feels unresolved.
>
> We agree with the feedback on the manuscript's flow. We will condense the philosophical framing in the introduction and add a brief callback in §5. This will effectively free up valuable space to include the new experiments detailed above in the main text.
>
> ### W2/Q3: Prior sphere-based approaches discussed too briefly.
>
> We agree. We will expand §2 to discuss the mechanics of prior sphere-based methods in detail: spherical CNNs (Zhao et al. 2019/2021), surface-based convolutional decoders (Gu et al. 2023), icosahedral patch-based transformers (SIM, Dahan et al. 2022/2024/2025), and Yu et al. (2025), as well as classical inter-subject registration pipelines (Fischl 2012; Robinson et al. 2014/2018; Glasser et al. 2016). We will explicitly state that our contribution is not introducing geometry-aware fMRI modelling per se, but rather *how* geometry and anatomy are operationalised: task-aligned ROI tokenization (SRST) combined with anatomy-conditioned MoE routing (SG-MoE).
>
> ### W3/Q1: The $\mathcal O(T^2)\to\mathcal O(T'^2)$ complexity reduction claim.
>
> The reviewer is right, our approach does not change the asymptotic complexity class, which remains quadratic. What changes is the input size: SRST reduces the token count from $T$=81,924 (full cortex) to $T'$=9,488 (visual ROI), an 88.4% reduction. We will fix this in the revision in the manuscript to explicitly distinguish between the unchanged theoretical complexity class and the massive reduction in practical computational burden.
>
> ### W4/Q2: Cortical features may act as implicit subject fingerprints.
>
> We appreciate the reviewer for raising this central concern. To definitively verify whether the model relies on structural geometry or merely uses anatomy as an implicit subject ID, we ran two new ablations, and compared them against our existing **Subject-ID MoE baseline** (Table 2, #2).
>
> **(1) Anatomical-swap experiment:** To directly address your concern, we replaced each subject's cortical maps with those of a different subject (1257 $\to$ 2571) while keeping the fMRI activity unchanged. **(2) Ablation without anatomy:** We remove structural input entirely and route using only fMRI signals. This tests whether anatomy helps at all beyond what functional data already provides.
>
> The results reveal a clear and consistent monotonic degradation across all metrics (e.g., on CLIP) when anatomical priors are progressively compromised or removed: **Full SG-MoE (Correct anatomy):** 93.2%; **Subject-ID MoE (Explicit ID gating):** 92.7%; **Anatomical Swap (Shuffled anatomy):** 91.9%; **No Anatomy:** 90.2% (Distance metrics follow the exact same degradation pattern: SwAV worsens from 0.404  $\to$ 0.407 $\to$ 0.411 $\to$ 0.417).
>
> This strict performance ordering strongly counteracts the memorization hypothesis. If the anatomical features were functioning merely as implicit "patient IDs", we would expect an explicit one-hot Subject-ID to perform optimally, and a swapped anatomy (which still provides a unique identifier) to perform comparably.
>
> Instead, we observe that routing with correct anatomical features outperforms explicit ID routing (93.2% vs. 92.7%), demonstrating that anatomy provides rich geometric priors beyond mere identity. Furthermore, introducing mismatched anatomy actively disrupts the semantic alignment (dropping to 91.9%), and removing it entirely results in a severe 3-point drop (90.2%). Together, these findings confirm that the model's performance gains stem from genuine, biologically meaningful structure-function coupling rather than arbitrary identity tagging.
>
> **Balanced feature importance:** We also agree with your observation regarding the balanced weights of highly variable features, but we clarify that this is actually a desirable outcome. The four features capture complementary morphological dimensions. Their balanced importance indicates that the MoE router has learned a robust combinatorial rule. Had a single feature heavily dominated, it would indeed be highly suspicious of acting as a fingerprint. Your observation helped us recognize the need to articulate this crucial distinction more clearly, which we will add to the final manuscript.

---

> > ### Author Rebuttal · Reviewer_Ax4V · 2026-04-02
> >
> > Thanks to the authors for their clarification and for their planned edits.
> > The extensive experiments provided and planned clarifications for the final version of the paper have addressed my concerns; I will raise my score to 5.

---

> > > ### Author Response · Authors · 2026-04-02
> > >
> > > We sincerely thank the reviewer for the re-evaluation and for the constructive feedback throughout the review process. We will ensure all planned edits are incorporated in the camera-ready version!

---

### Decision · Program_Chairs · 2026-04-30

**Decision:**

Accept (regular)

**Comment:**

This paper tackles the problem of neural fMRI-based brain decoding. The main contribution of this paper is architectural : the spherical tokenizer, and the mixture-of-expert model. The experiments are done at large scale and the results are strong. All reviewers salute the strength of the empirical evidence at scale.
The proposed architecture is novel (at least in this space) and the proposed inductive bias works very well with the domain. The experiments are thorough, with good ablation studies and visualizations.
Reviewers have pointed about potential implicit subject identification through cortical features and a gap between the theoretical motivation and implementation. The authors' rebuttal, including anatomy-swap ablations showing monotonic performance degradation, largely addressed these issues, and no reviewer maintained major objections.
For all the above reasons, I recommend this paper for acceptance. Great work - congrats!